# Leveraging sparse and shared feature activations for disentangled representation learning

**Marco Fumero**
Sapienza, University of Rome

**Florian Wenzel**
Amazon AWS

**Luca Zancato**
Amazon AWS

**Alessandro Achille**
Amazon AWS

**Emanuele Rodolà**
Sapienza, University of Rome

**Stefano Soatto**
Amazon AWS

**Bernhard Schölkopf**
Amazon AWS

**Francesco Locatello**
IST Austria

## Abstract

Research on recovering the latent factors of variation of high dimensional data has so far focused on simple synthetic settings. Mostly building on unsupervised and weakly-supervised objectives, prior work missed out on the positive implications for representation learning on real world data. In this work, we propose to leverage knowledge extracted from a diversified set of supervised tasks to learn a common disentangled representation. Assuming that each supervised task only depends on an unknown subset of the factors of variation, we disentangle the feature space of a supervised multi-task model, with features activating sparsely across different tasks and information being shared as appropriate. Importantly, we never directly observe the factors of variations, but establish that access to multiple tasks is sufficient for identifiability under sufficiency and minimality assumptions. We validate our approach on six real world distribution shift benchmarks, and different data modalities (images, text), demonstrating how disentangled representations can be transferred to real settings.

## 1 Introduction

A fundamental question in deep learning is how to learn meaningful and reusable representation from high dimensional data observations [8, 75, 78, 77]. A core area of research pursuing is centered on disentangled representation learning (DRL) [56, 8, 33] where the aim is to learn a representation which recovers the factors of variations (FOVs) underlying the data distribution. Disentangled representations are expected to contain all the information present in the data in a compact and interpretable structure [46, 16] while being independent from a particular task [29]. It has been argued that separating information into interventionally independent factors [78] can enable robust downstream predictions, which was partially validated in synthetic settings [19, 58]. Unfortunately, these benefits did not materialize in real world representations learning problems, largely limited by a lack of scalability of existing approaches.

In this work we focus on leveraging knowledge from different task objectives to learn better representations of high dimensional data, and explore the link with disentanglement and out-of-distribution (OOD) generalization on real data distributions. Representations learned from a large diversity of tasks are indeed expected to be richer and generalize better to new, possibly out-of-distribution, tasks. However, this is not always the case, as different tasks can compete with each other and lead to weaker models. This phenomenon, known as negative transfer [61, 91] in the context of transfer learning or task competition [83] in multitask learning, happens when a limited capacity model is used to learn two different tasks that require expressing high feature variability and/or coverage. Aiming to use the same features for different objectives makes them noisy and often increases the sensitivity to spurious correlations [35, 27, 7], as features can be both predictive and detrimental for

37th Conference on Neural Information Processing Systems (NeurIPS 2023).

different tasks. Instead, we leverage a diverse set of tasks and assume that each task only depends on an unknown subset of the factors of variation. We show that disentangled representations naturally emerge without any annotation of the factors of variations under the following two representation constraints:

- *Sparse sufficiency*: Features should activate sparsely with respect to tasks. The representation is *sparsely sufficient* in the sense that any given task can be solved using few features.
- *Minimality*: Features are maximally shared across tasks whenever possible. The representation is *minimal* in the sense that features are encouraged to be reused, i.e., duplicated or split features are avoided.

These properties are intuitively desirable to obtain features that (i) are disentangled w.r.t. to the factors of variations underlying the task data distribution (which we also theoretically argue in Proposition 2.1), (ii) generalize better in settings where test data undergo distribution shifts with respect to the training distributions, and (iii) suffer less from problems related to negative transfer phenomena. To learn such representations in practice, we implement a meta learning approach, enforcing feature sufficiency and sharing with a *sparsity* regularizer and an entropy based *feature sharing* regularizer, respectively, incorporated in the base learner. Experimentally, we show that our model learns meaningful disentangled representations that enable strong generalization on real world data sets. Our contributions can be summarized as follows:

- We demonstrate that is possible to learn disentangled representations leveraging knowledge from a distribution of tasks. For this, we propose a meta learning approach to learn a feature space from a collection of tasks while incorporating our sparse sufficiency and minimality principles favoring task specific features to coexist with general features.
- Following previous literature, we test our approach on synthetic data, validating in an idealized controlled setting that our sufficiency and minimality principles lead to disentangled features w.r.t. the ground truth factors of variation, as expected from our identifiability result in Proposition 2.1.
- We extend our empirical evaluation to non-synthetic data where factors of variations are not known, and show that our approach generalizes well out-of-distribution on different domain generalization and distribution shift benchmarks.

## 2 Method

Given a distribution of tasks $t \sim \mathcal{T}$ and data $(\mathbf{x_t}, y_t) \sim \mathcal{P}_t$ for each task $t$, we aim to learn a disentangled representation $g(\mathbf{x}) = \hat{\mathbf{z}} \in \hat{\mathcal{Z}} \subseteq \mathbb{R}^M$, which generalizes well to unseen tasks. We learn this representation $g$ by imposing the sparse sufficiency and minimality inductive biases.

### 2.1 Learning sparse and shared features

Our architecture (see Figure 1) is composed of a backbone module $g_\theta$ that is shared across all tasks and a separate linear classification head $f_{\phi_t}$, which is specific to each task $t$. The backbone is responsible to compute and learn a general feature representation for all classification tasks. The linear head solves a specific classification problem for the task-specific data $(\mathbf{x_t}, y_t) \sim \mathcal{P}_t$ in the feature space $\hat{\mathcal{Z}}$ while enforcing the feature sufficiency and minimality principles. Adopting the typical meta-learning setting [34], the backbone module $g_\theta$ can be viewed as the *meta learner* while the task-specific classification heads $f_{\phi_t}$ can be viewed as the *base learners*. In the meta-learning setting we assume to have access to samples for a new task given by a *support set* $U$, with elements $(\mathbf{x}^U, y^U) \in U$. These samples are used to fit the linear head $f_{\phi^*}$ leading to the optimal feature weights for the given task. For a *query* $\mathbf{x}^Q \in Q$, the prediction is obtained by computing the forward pass $\hat{y} = f_{\phi^*}(g_\theta(\mathbf{x}^Q))$.

**Enforcing feature minimality and sufficiency.** To solve a task in the feature space $\hat{\mathcal{Z}}$ of the backbone module we impose the following regularizer $Reg(\phi)$ on the classification heads $f_\phi$ with parameter $\phi \in \mathbb{R}^{T \times M \times C}$, where $T$ is the number of tasks, $M$ the number of features, and $C$ the number of classes. The regularizer is responsible for enforcing the feature minimality and sufficiency

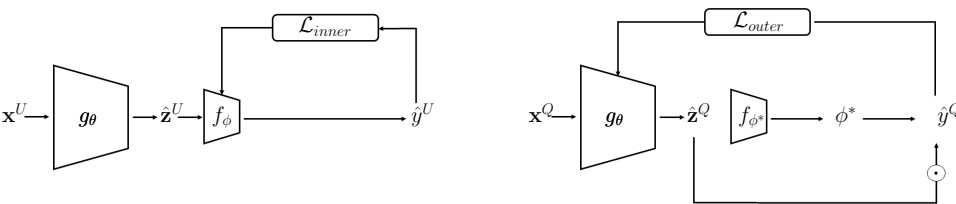

Figure 1: *Model scheme*: Illustrations of the (*Top*) the inner loop stage and outer loop following the steps of the algorithmic procedure described in Section B.1 in the Appendix.

properties. It is composed of the weighted sum of a sparsity penalty $Reg_{L1}$ and an entropy-based feature sharing penalty: $Reg_{sharing}$

$$Reg(\phi) = \alpha Reg_{L_1}(\phi) + \beta Reg_{sharing}(\phi), \tag{1}$$

with scalar weights $\alpha$ and $\beta$. The penalty terms are defined by:

$$Reg_{L_1}(\phi) = \frac{1}{TC} \sum_{t,c,m} |\phi_{t,m,c}| \tag{2}$$

$$Reg_{sharing}(\phi) = H(\tilde{\phi}_m) = - \sum_m \tilde{\phi}_m log(\tilde{\phi}_m) \tag{3}$$

where $\tilde{\phi}_m = \frac{1}{TC} \frac{\sum_{t,c} |\phi_{t,c,m}|}{\sum_{t,c,m} |\phi_{t,c,m}|}$ are the normalized classifier parameters. Sufficiency is enforced by a sparsity regularizer given by the $L_1$-norm, which constrains classification head to use only a sparse subset of the features. Minimality is enforced by the feature sharing term: minimizing the entropy of the distribution of feature importances (i.e. normalized $|\phi_t|$) averaged across a mini batch of $T$ tasks, leads to a more peaked distribution of activations across tasks. This forces features to cluster across tasks and therefore be reused by different tasks, when useful. We remark that different choices for the regularizers coming from the linear multitask learning literature (e.g. [59, 39, 38]) to enforce sparse sufficiency and minimality are indeed possibile. We leave their exploration as a future direction.

## 2.2 Training method

We train the model in meta-learning fashion by minimizing the test error over the expectation of the task distribution $t \sim \mathcal{T}$. This can be formalized as a *bi-level optimization problem*. The optimal backbone model $g_{\theta*}$ is given by the *outer optimization problem*:

$$\min_\theta \mathbb{E}_t [\mathcal{L}_{outer}(f_{\phi*}(g_\theta(\mathbf{x}_t^Q), y_t^Q))], \tag{4}$$

where $f_{\phi*}$ are the optimal classifiers obtained from solving the *inner optimization problem*, and $(\mathbf{x}_t^Q, y_t^Q) \in Q_t$ are the test (or query) datum from the query set $Q_t$ for task $t$. Let $U_t$ be the support set with samples $(\mathbf{x}_t^U, y_t^U) \in U$ for task $t$, where typically the support set is distinct from the query set, i.e., $U \cap Q = \emptyset$. The optimal classifiers $f_{\phi*}$ are given by the *inner optimization problem*:

$$\min_\phi \frac{1}{T} \sum_t \mathcal{L}_{inner}(\hat{y}_t^U, y_t^U) + Reg(\phi), \tag{5}$$

where $\hat{y}_t^U = f_\phi(g_\theta(\mathbf{x}_t^U))$. For both the inner loss $\mathcal{L}_{inner}$ and outer loss $\mathcal{L}_{outer}$ we use the cross entropy loss.

**Task generation.** Our method can be applied in a standard supervised classification setting where we construct the tasks on the fly as follows. We define a task $t$ as a $C$-way classification problem. We first select a random subset of $C$ classes from a training domain $D_{train}$ which contains $K_{train}$ classes. For each class we consider the corresponding data points and select a random support set $U_t$ with elements $(\mathbf{x}_t^U, y^U) \in U$ and a disjoint random query set $Q_t$ with elements $(\mathbf{x}_t^Q, y^Q) \in Q_t$.

**Algorithm.** In practice we solve the bi-level optimization problem (4) and (5) as follows. In each iteration we sample a batch of $T$ tasks with the associated support and query set as described above. First, we use the samples from the support set $S_t$ to fit the linear heads $f_\phi$ by solving the inner optimization problem (5) using stochastic gradient descent for a fixed number of steps. Second, we

use the samples from the query set $Q_t$ to update the backbone $g_\theta$ by solving the outer optimization problem (4) using implicit differentiation [11, 31]. Since the optimal solution of the linear heads $\phi^*$ depend on the backbone $g_\theta$, a straightforward differentiation w.r.t. $\theta$ is not possible. We remedy this issue by using the approximation strategy of [28] to compute the implicit gradients. The algorithm is summarized in section B.1 of the Appendix.

## 2.3 Theoretical analysis

We analyze the implications of the proposed minimality and sparse sufficiency principles and show in a controlled setting that they indeed lead to identifiability. As outlined in Figure 2, we assume that there exists a set of independent latent factors $\mathbf{z} \sim \prod_{i=1}^{d} p(z_i)$ that generate the observations via an unknown mixing function $\mathbf{x} = g^*(\mathbf{z})$. Additionally, we assume that the labels $y_t$ for a task $t$ only depend on a subset of the factors indexed by $S_t \sim P(S)$, where $S$ is an index set on $\mathbf{z} \in \mathcal{Z}$, via some unknown mixing function $y_t = f_t^*(\mathbf{z})$ (potentially different for different tasks). We formalize the two principles that are imposed on $f^*$ by:

1. *sufficiency*: $f_t^* = f_t^*|_{S_t}$ for $S_t \sim p(\mathcal{S})$
2. *minimality*: $\nexists S' \neq S_t \subset \mathcal{S}$ s.t. $f_t^*|_{S'} = f_t^*$,

where $f|_{S_t}$ denotes that the input to a function $f$ is restricted to the index set given by $S_t$ (all remaining entries are set to zero). (1) states that $f_t^*$ only uses a subset of features, and (2) states that there are not be duplicate features.

**Proposition 2.1.** *Assume that $g^*$ is a diffeomorphism (smooth with smooth inverse), $f^*$ satisfies the sufficiency and minimality properties stated above, and $p(S)$ satisfies: $p(S \cap S' = \{i\}) > 0$ or $p(\{i\} \in (S \cup S') - (S' \cap S)) > 0$. Observing unlimited data from $p(X, Y)$, it is possible to recover a representation $\hat{\mathbf{z}}$ that is an axis aligned, component wise transformation of $\mathbf{z}$.*

**Remarks:** Overall, we see this proposition as validation that in an idealized setting our inductive biases are sufficient to recover the factors of variation. Note that the proof is non-constructive and does not entail a specific method. In practice, we rely on the same constraints as inductive biases that lead to this theoretical identifiability and experimentally show that disentangled representations emerge in controlled synthetic settings. On real data, (1) we cannot directly measure disentanglement, (2) a notion of global ground-truth factors may even be ill-posed, and (3) the assumptions of Proposition 2.1 are likely violated. Still, sparse sufficiency and minimality yield some meaningful factorization of the representation for the considered tasks.

**Relation to [47] and [58]**: Our theoretical result can be reconnected with concurrent work [47] and can be seen as a corollary with a different proof technique and slightly relaxed assumptions. The main difference is that our feature minimality allows us to also cover the case where the number of factors of variations is unknown, which we found critical in real world data sets (the main focus of our paper). Instead, they only assume sparse sufficiency, which is enough for identifiability if the ground-truth number of factors is known, but is not enough to recover high disentaglement when this is not the case (see Figure 3) and does not translate well to real data, see Table 16 with the empirical comparison in Appendix D.8. Interestingly, their analysis also hints at the fact that our approach also benefits in terms of sample complexity on transfer learning downstream tasks. Our proof technique follows the general construction developed for multi-view data in [58], adapted to our different setting. Instead of observing multiple views with shared factors of variation, we observe a single task that only depend on a subset of the factors.

## 3 Related work

**Learning from multiple tasks and domains.** Our method addresses the problem of learning a general representation across multiple and possibly unseen tasks [15, 103] and environments [105, 32, 44, 97, 63, 94, 64] that may be competing with each other during training [61, 91, 83]. Prior research tackled task competition by introducing task specific modules that do not interact during training [67, 101, 80]. While successfully learning specialized modules, these approaches can not leverage synergistic information between tasks, when present. On the other hand, our approach is closer to multi-task methods that aim at learning a generalist model, leveraging multi-task interactions [106, 5]. Other approaches that leverage a meta-learning objective for multi-task learning have been

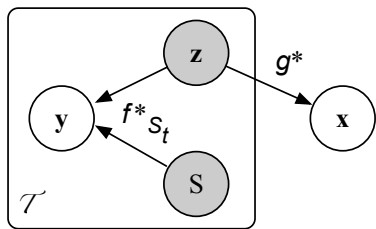

Figure 2: Assumed causal generative model: the gray variables are unobserved. Observations $\mathbf{x}$ are generated by some unknown mixing of a set of factors of variations $\mathbf{z}$. Additionally, we observe a distribution of supervised tasks, only depending on a subset of factors of variations indexed by $S$.

formulated [18, 81, 50, 9]. In particular, [50] proposes to learn a generalist model in a few-shot learning setting without explicitly favoring feature sharing, nor sparsity. Instead, we rephrase the multi-task objective function encoding both feature sharing and sparsity to avoid task competition.

Similar to prior work in domain generalization, we assume the existence of stable features for a given task [64, 4, 86, 40, 90] and amortize the learning over the multiple environments. Differently than prior work, we do not aim to learn an invariant representation a priori. Instead, we learn sufficient and minimal features for each task, which are selected at test time fitting the linear head on them. In light of [32], one can interpret our approach as learning the final classifier using empirical risk minimization but over features learned with information from the multiple domains.

**Disentangled representations.** Disentanglement representation learning [8, 33] aims at recovering the factors of variations underlying a given data distribution. [56] proved that without any form of supervision (whether direct or indirect) on the Factors of Variation (FOV) is not possible to recover them. Much work has then focused on identifiable settings [58, 25] from non-i.i.d. data, even allowing for latent causal relations between the factors. Different approaches can be largely grouped in two categories. First, data may be non-independently sampled, for example assuming sparse interventions or a sparse latent dynamics [30, 55, 13, 100, 2, 79, 48]. Second, data may be non-identically distributed, for example being clustered in annotated groups [37, 41, 82, 95, 60]. Our method follows the latter, but we do not make assumptions on the factor distribution across tasks (only their relevance in terms of sufficiency and minimality). This is also reflected in our method, as we train for supervised classification as opposed to contrastive or unsupervised learning as common in the disentanglement literature. The only exception is the work of [47] discussed in Section 2.3.

## 4 Experiments

We start by highlighting here the experimental setup of this paper along with its motivation.

**Synthetic experiments.** We first evaluate our method on benchmarks from the disentanglement literature [62, 14, 71, 49] where we have access to ground-truth annotations and we can assess quantitatively how well we can learn disentangled representations. We further investigate how minimality and feature sharing are correlated with disentanglement measures (Section 4.1) and how well our representations, which are learned from a limited set of tasks, generalize their composition. The purpose of these experiments is to validate our theoretical statement, showing that if the assumptions of Proposition 2.1 hold, our methods quantitatively recover the factors of variation.

**Domain generalization.** On real data sets, we can neither quantitatively measure disentanglement nor are we guaranteed identifiability (as assumptions may be violated). Ultimately, the goal of disentangled representations is to learn features that lend themselves to be easily and robustly transferred to downstream tasks. Therefore, we first evaluate the usefulness of our representations with respect to downstream tasks subject to distribution shifts, where isolating spurious features was found to improve generalization in synthetic settings [19, 58] To assess how robust our representations are to distribution shifts, we evaluate our method on domain generalization and domain shift tasks on six different benchmarks (Section 4.2). In a domain generalization setting, we do not have access to samples coming from the testing domain, which is considered to be OOD w.r.t. to the training domains. However, in order to solve a new task, our method relies on a set labeled data at test time to fit the linear head on top of the feature space. Our strategy is to sample data points from the training

distribution, balanced by class, assuming that the label set $Y$ does not change in the testing domain, although its distribution may undergo subpopulation shifts.

**Few-shot transfer learning.** Lastly, we test the adaptability of the feature space to new domains with limited labeled samples. For transfer learning tasks, we fit a linear head using the available limited supervised data. The sparsity penalty $\alpha$ is set to the value used in training; the feature sharing parameter $\beta$ is defaulted to zero unless specified.

**Experimental setting.** To have a fair comparison with other methods in the literature, we adopt the standard experimental setting of prior work [32, 44]. Hyperparameters $\alpha$ and $\beta$ are tuned performing model selection on validation set, unless specified otherwise. For comparison with baselines, we substitute our backbone with that of the baseline (e.g. for ERM models, we detach the classification head) and then fit a new linear head on the same data. The linear head module trained at test time on top of the features is the same both for our and compared methods. Despite its simplicity, we report the ERM baseline for comparison in our experiments in the main paper, since it has been shown to perform best in average on domain generalization benchmarks [32, 44]. We further compare with other consolidated approaches in the literature such as IRM [4], CORAL [85] and GroupDRO [73] and include a large and comprehensive comparison with [99, 10, 51, 53, 26, 54, 65, 102, 36, 45] in AppendixD.4. Experimental details are fully described in Appendix C.

## 4.1 Synthetic experiments

We start by demonstrating that our approach is able to recover the factors of variation underlying a synthetic data distribution like [62]. For these experiments, we assume to have partial information on a subset of factors of variation $Z$, and we aim to learn a representation $\hat{\mathbf{z}}$ that aligns with them while ignoring any spurious factors that may be present. We sample random tasks from a distribution $\mathcal{T}$ (see Appendix C.3 for details) 5and focus on binary tasks, with $Y = \{0, 1\}$. For the `DSprites` dataset an example of valid task is *"There is a big object on the left of the image"*. In this case, the partially observed factors (quantized to only two values) are the *x position* and *size*. In Table 1, we show how the feature sufficiency and minimality properties enable disentanglement in the learned representations. We train two identical models on a random distribution of sparse tasks defined on FOVs, showing that, for different datasets [62, 14, 49, 71], the same model without regularizers achieves a similar in-distribution (ID) accuracy, but a much lower disentanglement.

We then randomly draw and fix 2 groups of tasks with supports $S_1, S_2$ (18 in total), which all have support on two FOVs, $|S_1| = |S_2| = 2$. The groups share one factor of variation and differ in the other one, i.e. $S_1 \cap S_2 = \{i\}$ for some $\{i\} \in Z$. The data in these tasks are subject to spurious correlations, i.e. FOVs not in the task support may be spuriously correlated with the task label. We start from an overestimate of the dimension of $\tilde{\mathbf{z}}$ of 6, trying to recover $\mathbf{z}$ of size 3. We train our network to solve these tasks, enforcing sufficiency and minimality on the representation with different regularization degrees. In Figure 3, we show how the alignment of the learned features with the ground truth factors of variations depend on the choice of $\alpha, \beta$, going from no disentanglement ($DCI = 27.8$). to good alignment as we enforce more sufficiency and minimality. The model that attains the best alignment ($DCI = 98.8$) uses both sparsity and feature sharing. Sufficiency alone (akin to the method of [47]) is able to select the right support for each task, but features are split or duplicated, attaining lower disentanglement ($DCI = 71.9$). The feature sharing penalty ensures clustering

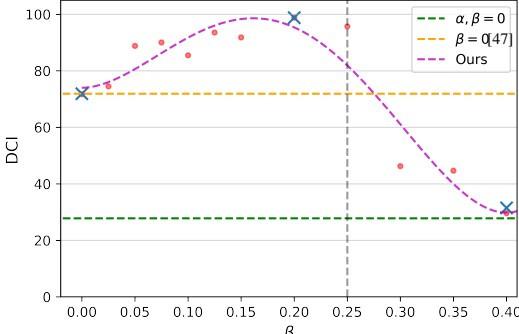

Figure 3: *Role of minimality*: We plot the DCI metric of a set of models (*red dots*) trained on fixed tasks from `DSprites`: Training without regularizers leads to no disentanglement (*green*). Enforcing sparsity alone (*yellow*, akin to [47]) achieves good disentanglement ($DCI = 71.9$), but some features may be split or duplicated. Enforcing both minimality and sparse sufficiency (*magenta*) attains the best $DCI$ (98.8). When $\beta$ is too high ($> 0.25$) activated features collapses into few clusters with respect to tasks. For complete results and experiments on additional datasets see Table 8 and Figures 6, 7 in Appendix.

in the feature space w.r.t. tasks, ensuring to reach high disentanglement, although it may result in the failure cases, when $\beta$ is too high ($\beta > 0.25$).

Table 1: *Enforcing disentanglement*: DCI [22] disentanglement scores and ID accuracy on test samples for a model trained without enforcing sufficiency and minimality (top row), and model with the regularizers activated (bottom row). While attaining similar performance on accuracy, the model with the activated regularizer always show higher disentanglement. See Table 7 for additional scores.

|  | Dsprites | 3Dshapes | SmallNorb | Cars |
|---|---|---|---|---|
| *No reg* (DCI,Acc) | (16.6,94.4) | (44.4,96.2 ) | (16.5,96.1) | (60.5,99.8) |
| $\alpha, \beta$ (DCI,Acc) | (**69.9**,95.8) | (**87.7**, 95.8) | (**55.8**,95.6 ) | (**92.3**,99.8 ) |

**Disentanglement and minimality are correlated.** In the synthetic setting, we also show the role of the feature sharing penalty. Minimizing the entropy of feature activations across mini-batches of tasks results in clusters in the feature space. We investigate how the strength of this penalty correlates well with disentanglement metrics [22] training different models on `Dsprites` which differ by the value of $\beta$. For 15 models trained increasing $\beta$ from 0 to 0.2 linearly, we observe a correlation coefficient with the DCI metric associated to representations compute by each model of 94.7, showing that the feature sharing property strongly encourages disentanglement. This confirms again that sufficiency alone (i.e. enforcing sparsity) is not enough to attain good disentanglement.

**Task compositional generalization.** Finally, we evaluate the generalization capabilities of the features learned by our method by testing our model on a set of unseen tasks obtained by combining tasks seen during training. To do

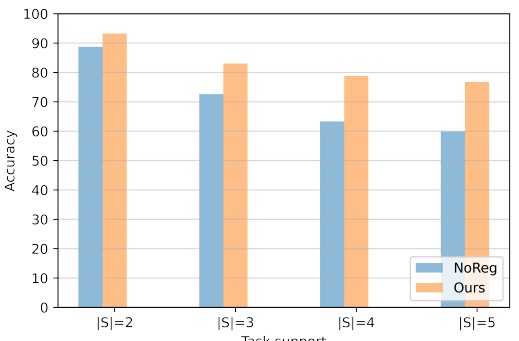

Figure 4: *Task compositional generalization*: Mean accuracy over 100 random test tasks reported for group of tasks of growing support (*second, third, fourth column*) for a model trained without inductive biases (*blue*, attaining $DCI = 29.4$) and enforcing them (*orange*, $DCI = 59.4$). The latter show better compositional generalization resulting from the properties enforced on the representation. Exact values are reported in Table 9 in Appendix.

this, we first train two models on the `AbstractDSprites` dataset using a random distribution of tasks, where we limit the support of each task to be within 2 (i.e. $|S| = 2$). The models differ in activating/deactivating the regularizers on the linear heads. Then, we test on 100 tasks drawn from a distribution with increasing support on the factors of variation ($|S| = 3, |S| = 4, |S| = 5$), which correspond to composition of tasks in the training distribution; see Figure 4, with the accompanying Table 9 in Appendix D.

## 4.2 Domain Generalization

In this section we evaluate our method on benchmarks coming from the domain generalization field [32, 93, 70] and subpopulation distribution shifts [73, 44], to show that a feature space learned with our inductive biases performs well out of real world data distribution.

**Subpopulation shifts.** Subpopulation shifts occur when the distribution of minority groups changes across domains. Our claim is that a feature space that satisfies sparse sufficiency and minimality is more robust to spurious correlations which may affect minority groups, and should transfer better to new distributions. To validate this, we test on two benchmarks `Waterbirds` [73], and `CivilComments` [44] (see Appendix C.1).

For both, we use the train and test split of the original dataset. In Table 4, last row, we report the results on the test set of `Waterbirds` for the different groups in the dataset (landbirds on land, landbirds on water, waterbirds on land, and waterbirds on water, respectively). We fit the linear head

Table 2: Quantitative results for few-shot transfer learning, with our method consistently outperforming ERM across all sample sizes and data sets.

| N-shot/Algorithm | OOD accuracy (averaged by domains) | | | |
|---|---|---|---|---|
| **1-shot** | PACS | VLCS | OfficeHome | Waterbirds |
| ERM | 80.5 | 59.7 | 56.4 | 79.8 |
| Ours | **81.5** | **68.2** | **58.4** | **88.4** |
| **5-shot** | | | | |
| ERM | 87.1 | 71.7 | 75.7 | 79.8 |
| Ours | **88.3** | **74.5** | **77.0** | **87.6** |
| **10-shot** | | | | |
| ERM | 87.9 | 74.0 | 81.0 | 84.2 |
| Ours | **90.4** | **77.3** | **82.0** | **89.2** |

Table 3: Quantitative evaluation on Camelyon17: we report accuracy both on ID and OOD splits. Our approach achieves significantly higher validation and test OOD accuracy.

| | Validation(ID) | Validation (OOD) | Test (OOD) |
|---|---|---|---|
| ERM | 93.2 | 84 | 70.3 |
| CORAL | **95.4** | 86.2 | 59.5 |
| IRM | 91.6 | 86.2 | 64.2 |
| Ours | 93.2 ±0.3 | **89.9**±0.6 | **74.1**±0.2 |

on a random subset of the training domain, balanced by class, repeat 10 times and report accuracy and standard deviation on test. For `CivilComments` we report the average and worst accuracy in Figure 5, where we compare with ERM and groupDRO [73]. While performing almost on par w.r.t. ERM, our method is more robust to spurious correlation in the dataset, showing the higher worst group accuracy. Importantly, we outperform GroupDRO, which uses information on the subdomain statistics, while we do not assume any prior knowledge about them. Results per group are reported in the Appendix (Table 11).

**DomainBed.** We evaluate the domain generalization performance on the `PACS`, `VLCS` and `OfficeHome` datasets from the DomainBed [32] test suite (see Appendix C.1 for more details). On these datasets, we train on $N - 1$ and leave one out for testing. Regularization parameters $\alpha$ and $\beta$ are tuned according to validation sets of `PACS`, and used accordingly on the other dataset. For these experiments we use a `ResNet50` pretrained on `Imagenet` [17] as a backbone, as done in [32] To fit the linear head we sample 10 times with different samples sizes from the training domains and we report the mean score and standard deviation. Results are reported in Table 4, showing how enforcing sparse sufficiency and minimality leads consistently to better OOD performance. Comparisons with 13 additional baselines is in Appendix D.4.

**Camelyon17.** The model is trained according to the original splits in the dataset. In Table 3 we report the accuracy of our model on in-distribution and OOD

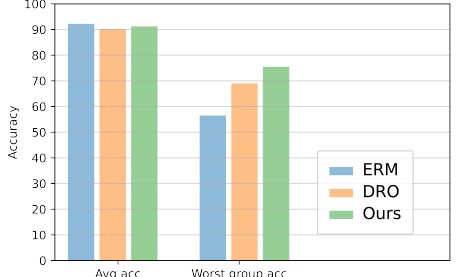

Figure 5: *Quantitative results on CivilComments*: we report the accuracy on test averaged across all demographic groups (*left group*), and the worst group accuracy, on the *right*. Our method (*green*) performs similarly in terms of average accuracy and outperforms in terms of worst group accuracy, without using any knowledge on the group composition in the training data. For exact values and error estimates, see Table 10 in the Appendix.

splits, compared with different baselines [84, 4]. Our method shows the best performance on the OOD test domains. The intuition of why this happens is that, due to minimality, we retain more features which are shared across the three training domains, giving less importance to the ones that are domain-specific (which contain the spurious correlations with the hospital environmental informations). This can be further enforced at test time, as we show in the ablation in Appendix D.9, trading off in distribution performance for OOD accuracy.

Table 4: Results for domain generalization on DomainBed. Our approach achieves consistently higher average OOD generalization, outperforming ERM in all cases except one.

| Dataset/Algorithm | OOD accuracy (by domain) | | | | |
|---|---|---|---|---|---|
| **PACS** | S | A | P | C | Average |
| ERM | $77.9 \pm 0.4$ | $\mathbf{88.1} \pm 0.1$ | $97.8 \pm 0.0$ | $79.1 \pm 0.9$ | 85.7 |
| Ours | $\mathbf{83.1} \pm 0.1$ | $86.7 \pm 0.8$ | $\mathbf{97.8} \pm 0.1$ | $\mathbf{83.5} \pm 0.1$ | **87.5** |
| **VLCS** | C | L | V | S | Average |
| ERM | $97.6 \pm 1.0$ | $63.3 \pm 0.9$ | $76.4 \pm 1.5$ | $72.2 \pm 0.5$ | 77.4 |
| Ours | $\mathbf{98.1} \pm 0.2$ | $\mathbf{63.4} \pm 0.5$ | $\mathbf{78.2} \pm 0.7$ | $\mathbf{73.9} \pm 0.8$ | **78.4** |
| **OfficeHome** | C | A | P | R | Average |
| ERM | $53.4 \pm 0.6$ | $62.7 \pm 1.1$ | $76.5 \pm 0.4$ | $77.3 \pm 0.$ | 67.5 |
| Ours | $\mathbf{56.3} \pm 0.1$ | $\mathbf{66.7} \pm 0.7$ | $\mathbf{79.2} \pm 0.5$ | $\mathbf{81.3} \pm 0.4$ | **70.9** |
| **Waterbirds** | LL | LW | WL | WW | Average |
| ERM | $98.6 \pm 0.3$ | $52.05 \pm 3$ | $68.5 \pm 3$ | $93 \pm 0.3$ | 81.3 |
| Ours | $\mathbf{99.5} \pm 0.1$ | $\mathbf{73.0} \pm 2.5$ | $\mathbf{85.0} \pm 2$ | $\mathbf{95.5} \pm 0.4$ | **90.5** |

### 4.3 Few-shot transfer learning.

We finally show the ability of features learned with our method to adapt to a new domain with a small number of samples in a few-shot setting. We compare the results with ERM in Table 2, averaged by domains in each benchmark dataset. The full scores for each domain are in Appendix D.5 for 1-shot, 5-shot, and 10-shot setting, reporting the mean accuracy and standard deviations over 100 draws. Our approach achieves consistently higher accuracy than ERM, showing the better adaptation capabilities of our minimal and sufficently sparse feature space.

### 4.4 Additional results

In Appendix D we report a large collection of additional results, including comparison with 14 baseline methods on the domain shift benchmarks (D.4), a qualitative and quantitative analysis on the minimality and sparse sufficiency properties in the real setting (D.2), a favorable additional comparison on meta learning benchmarks, with 6 other baselines including [47](D.8), an ablation study on the effect of clustering features at test time (D.9), and a demonstration on the possibility to obtain a task similarity measure as a consequence of our approach (D.7).

## 5 Conclusions

In this paper, we demonstrated how to learn disentangled representations from a distribution of tasks by enforcing feature sparsity and sharing. We have shown this setting is identifiable and have validated it experimentally in a synthetic and controlled setting. Additionally, we have empirically shown that these representations are beneficial for generalizing out-of-distribution in real-world settings, isolating spurious and domain specific factors that should not be used under distribution shift.

**Limitations and future work**: The main limitation of our work is the global assumption on the strength of the sparsity and feature sharing regularizers $\alpha$ and $\beta$ across all tasks. In real settings these properties of the representations might need to change for different tasks. We have already observed this in the synthetic setting in Figure 3, where when $\beta > 0.25$ features cluster excessively and are unable to achieve clear disentanglement and do not generalize well. Future work may exploit some level of knowledge on the task distribution (e.g. some measure of distance on tasks) in order to tune $\alpha, \beta$ adaptively during training, or to train conditioning on a distribution of regularization parameters as in [21], enabling more generalization at test time. Another limitation is in the sampling procedure to fit the linear head at test time: sampling randomly from the training set (balanced by class) may not be enough to achieve the best performance under distributions shifts. Alternative sampling procedures, e.g. ones that incorporate knowledge on the distribution shift if available (as in [43]), may lead to better performance at test time.

## Acknowledgments and Disclosure of Funding

Marco Fumero and Emanuele Rodolà were supported by the ERC grant no.802554 (SPECGEO), PRIN 2020 project no.2020TA3K9N (LEGO.AI), and PNRR MUR project PE0000013-FAIR. Marco Fumero and Francesco Locatello were partially at Amazon while working at this project. We thank Julius von Kügelgen, Sebastian Lachapelle and the anonymous reviewers for their feedback and suggestions.

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
