# A  Proof of Proposition 1

To prove Proposition 2.1 we rely on the same proof construction of [58], adapting it to our setting. Intuitively, the proposition states that when minimality and sparse sufficiency properties hold it is possible to recover the factors of variations $z$ given enough observations from $p(x, y)$, if the following assumptions on the task distribution hold: (i) the probability of two arbitrary tasks having a singleton intersection of support on the factor of variations is non zero; (ii) the probability that their difference of supports is a singleton is non zero.

The proof is sketched in three steps:

- First, we prove identifiability when the support $S$ of a task is arbitrary but fixed, where we drop the subscript $t$ for convenience.
- Second, we randomize on $S$, to extend the proof for $S$ drawn at random.
- Third, we extend the proof to the case when the dimensionality of $\mathcal{Z}$ is unknown and we start on overestimate of it to recover it.

**Identifiability with fixed task support** We assume the existence of the generative model in Figure 2, which we report here for convenience:

$$p(\mathbf{z}) = \prod_i p(z_i) \qquad\qquad S \sim p(S) \qquad\qquad (6)$$

$$\mathbf{x} = g^*(\mathbf{z}) \qquad\qquad y = f_S^*(\mathbf{z}) \qquad\qquad (7)$$

together with the assumptions specified in theorem statement. We fix the support of the task $S$. We indicate with $g : Z \to X$ the invertible smooth, candidate function we are going to consider, whose inverse corresponds to $q(\mathbf{z}|\mathbf{x})$. We denote with $T \in S$ which indexes the coordinate subspace of image of $g^{-1}$ corresponding to the unknown coordinate subspace $S$ of factors of variation on which the fixed task depends on. Fixing $T$ requires knowledge of $|S|$. The candidate function $g^{-1}$ must satisfy:

$$f|_T(g^{-1}(\mathbf{x})) = y \qquad\qquad (8)$$

$$f|_{\bar{T}}(g^{-1}(\mathbf{x})) \neq y \qquad\qquad (9)$$

where $\bar{T}$ denotes the indices in the complement of $T$. $f$ denotes a predictor which satisfies the same assumptions on $f^*$ on $T$. We parametrize $g^{-1}$ with $g^{*-1}$ and set:

$g^{-1} = h^{-1} \circ g^{*-1}$ where $h : [0, 1]^d \to Z$, mapping from the uniform distribution on $\mathbb{R}^d$ to $Z$. We can rewrite the two above constraints as:

$$f|_T(h^{-1}(z)) = y \qquad\qquad (10)$$

$$f|_{\bar{T}}(h^{-1}(z)) \neq y \qquad\qquad (11)$$

We claim that the only admissible functions $h^{-1}$ maps each entry in $\mathbf{z}$ to unique coordinate in $T$. We observe that due to its smoothness and invertibility, $h^{-1}$ maps $Z$ to the submanifolds $\mathcal{M}_s, \mathcal{M}_{\bar{s}}$, which are disjoint. By contradiction:

- if $\mathcal{M}_{\bar{S}}$ does not lie in $\bar{T}$ then minimality is violated.
- if $\mathcal{M}_S$ does not lie in $T$ then sufficiency is violated

$h^{-1}$ maps each entry in $\mathbf{z}$ to unique coordinate in $T$. Therefore there exist a permutation $\pi$ s.t.:

$$h_T^{-1}(\mathbf{z}) = \bar{h}_T(\mathbf{z}_{\pi(S)}) \qquad\qquad (12)$$

$$h_{\bar{T}}^{-1}(\mathbf{z}) = \bar{h}_{\bar{T}}(\mathbf{z}_{\pi(\bar{S})}) \qquad\qquad (13)$$

The Jacobian of $h^{-1}$ is a blockwise matrix with block indexed by $T$. So we can identify the two blocks of factors in $S, \bar{S}$ but not necessarily the factors within, as they may be still entangled.

**Randomization on $S$**

we now consider $S$ to be drawn at random, therefore we observe $p(\mathbf{x}, y|S)$ without never observing $S$ directly. $g^{-1}$ must now associate each $p(\mathbf{x}, y)$ with a unique $T$, as well as a unique predictor $f$, for each $S \sim p(S)$ Indeed suppose that $p(\mathbf{x}, y|S = S_1)$ and $p(\mathbf{x}, y|S = S_2)$ with $S_1, S_2 \sim p(S)$ and $S_1 \neq S_2$. Then if $T$ would be the same for both tasks (as $f$), eq (6) could only be satisfied for a subset of size $|S_1 \cap S_2| < |S_1 \cup S_2|$, while $T$ is required to be of size $|S_1 \cup S_2|$ This corresponds to say that each task has its own sparse support and its own predictor. Conversely all $p(\mathbf{x}, y) \in supp(p(\mathbf{x}, y|S))$ need to be associated to the $T$ and the same predictor $f$, since they will all share the same subspace and cannot be associated to different $T$. Notice also that $|S_1 \cap S_2| = |T_1 \cap T_2|$ and $|S_1 \cup S_2| = |T_1 \cup T_2|$. We further assume:

$\forall z_i$ either $p(S \cap S' = \{i\}) > 0$ or $p(\{i\} \in (S \cup S') - (S' \cap S)) > 0$

We observe every factor as the intersection of the sets $S, S'$ which will be reflected in $T, T'$ or we observe single factors in the difference between the intersection and the union of $S, S'$. Examples of the two cases are illustrated below:

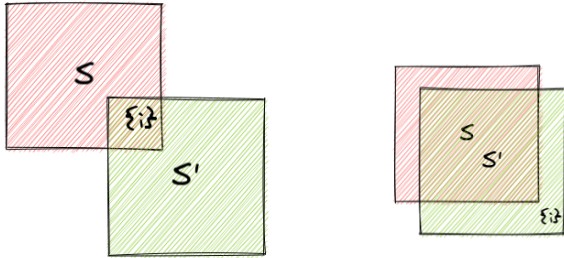

This together with (8) and (9) implies:

$$h_i^{-1}(\mathbf{z}) = \bar{h}_i(z_{\pi(i)}) \quad \forall i \in [d] \tag{14}$$

This further implies that the jacobian of $\bar{h}$ is diagonal. By the change of variable formula we have:

$$q(\hat{\mathbf{z}}) = p(\tilde{h}(\mathbf{z}_{\pi([d])})) \left| det \frac{\partial}{\partial \mathbf{z}_{\pi([d]))}} \tilde{h} \right| = \prod_{i01}^{d} p(\tilde{h}_i(z_{\pi(i)})) \left| \frac{\partial}{\partial z_{\pi(i)}} \tilde{h}_i \right| \tag{15}$$

This holds for the jacobian being diagonal and invertibility of $\tilde{h}$. Therefore $q(\hat{\mathbf{z}})$ is a coordinate-wise reparametrization of $p(\mathbf{z})$ up to a permutation of the indices. A change in a coordinate of $\mathbf{z}$ implies a change in the unique corresponding coordinate of $\hat{\mathbf{z}}$, so $g$ disentangles the factors of variation.

**Dimensionality of the support $S$**

Previously we assumed that the dimension of $\hat{\mathbf{z}}$ is the same as $\mathbf{z}$. We demonstrate that even when $d$ is unknown starting from an overstimate of it, we can still recover the factors of variations. Specifically, we consider the case when $\hat{d} > d$. In this case our assumption about the invertibility of $h$ is violated. We must instead ensure that $h$ maps $Z$ to a subspace of $\hat{Z}$ with dimension $d$. To substitute our assumption on inveribility on $h$, we will instead assume that $\mathbf{z}$ and $\hat{\mathbf{z}}$ have the same mutual information with respect to task labels $Y$, i.e.$I(Z, Y) = I(\hat{Z}, Y)$ Note that mutual information is invariant to invertible transformation, so this property was also valid in our previous assumption.

Now, consider two arbitrary tasks with $|S \cap S'| \neq \emptyset = k$ but $|T \cap T'| < k$, i.e. some features are duplicated/splitted. Hence $f, f'$ while have different support , i.e.:

$$f|_T = f'|_{T'} = f^*$$

We observe that in this situation nor sufficiency, nor minimality are necessarily violated because:

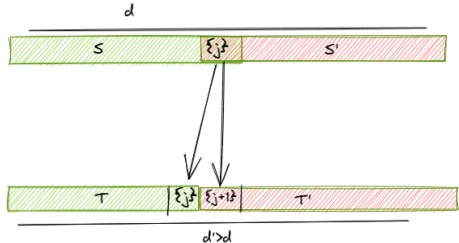

- $f|_T = f'|_{T'} = f^*$ (sufficiency is not violated)

- $T \cap T' = \emptyset \implies T \not\subset T', T' \not\subset T$ (minimality is not violated)

In other words we must ensure that a single fov $z_i$ is not mapped to different entries in $\hat{\mathbf{z}}$ (feature splitting or duplication). We fix two arbitrary tasks with $|S \cap S'| \neq \emptyset = k$ but $|T \cap T'| < k$, i.e. some features are duplicated. We know that $|S| = |T|$ and $|S'| = |T'|$ otherwise sufficency and minimaliy would be violated. Then if $|T \cap T'| < k$, then $|T \cup T'| > |S \cup S'| = d - k$ we have $p(|T \cup T'|) = p(supp(p(y|\hat{\mathbf{z}})) + supp(p'(y'|\hat{\mathbf{z}}'))) = p(\sum_i supp(f_i(.)))$ , and since

$$H[p(\sum_i supp(f_i(.)))] > H[p(\sum_i supp(f_i(.)))] \tag{16}$$

but we have assumed:

$$I(Z, Y) = I(\hat{Z}, Y) \tag{17}$$

$$\cancel{H(Y)} - H(Y|\hat{Z}) = \cancel{H(Y)} - H(Y|Z) \tag{18}$$

$$H(Y|\hat{Z}) = H(Y|Z) \tag{19}$$

$$H[p(Y|\hat{Z}) > 0] = H[p(Y|Z) > 0] \tag{20}$$

$$2^{H[p(Y|\hat{Z})>0]} = 2^{H[p(Y|Z)>0]} \tag{21}$$

$$|supp(p(Y|\hat{Z}))| = |supp(p(Y|Z)| \tag{22}$$

this last passage is due to relation between cardinality and entropy: for uniform distributions the exponential of the entropy is equal to the cardinality of the support of the distribution.

$$|supp(f)| = |supp(f^*)| \tag{23}$$

We know that (12) must hold for every task, therefore: $\sum_i I(Z, Y_i) = \sum_i I(\hat{Z}, Y_i)$ for each $i$ then: $\sum_i |supp(\hat{f}_i)| = \sum_i |supp(f_i^*)| \, |\bigcup_i T_i| = |\bigcup_i S_i|$ therefore (12) contradicts our assumption (13).

# B  Implementation details

## B.1  Training algorithm

---

**Algorithm 1** Training algorithm

---

 1: **Input:** A task distribution $\mathcal{T}$
 2: **while** Not converged **do**
 3:     Sample a batch $B_T$ of $T$ tasks $t \sim \mathcal{T}$
 4:     Sample $(U_t, Q_t)$ from each task in the batch
 5:     # **Inner loop**
 6:     **for** each $t$ in $B_T$ **do**
 7:         Compute $\mathbf{z}_t^U = g_\theta(\mathbf{x}_t^U)$
 8:     **end for**
 9:     Solve $\phi^* = argmin_\phi \frac{1}{T} \sum_t \mathcal{L}_{inner}(f_\phi(\mathbf{z}_t^U), y_t^U) + Reg(\phi)$
10:     # **Outer loop**
11:     **for** each $t$ **do**
12:         Compute $\mathbf{z}_t^Q = g_\theta(\mathbf{x}_t^Q)$
13:     **end for**
14:     Compute $\mathcal{L}_{outer}(f_{\phi^*}(g_\theta(\mathbf{x}_t^Q), y_t^Q))$
15:     Compute $\frac{\partial \mathcal{L}_{outer}(\theta)}{\partial \theta}$ as in [28]
16:     Update $\theta$
17: **end while**

---

## B.2  Implicit gradients

In the backward pass, denoting with $\mathcal{L}_{outer}^* = \mathcal{L}_{outer}(f_\phi^*(g_\theta(x^Q)), Y^Q)$ denoting the loss computed with respect to the optimal classifier $f_\phi^*$ on the query samples $(x^Q, Y^Q)$, we have to compute the following gradient:

$$\frac{\partial \mathcal{L}_{outer}^*(\theta)}{\partial \theta} = \frac{\partial \mathcal{L}_{outer}(\theta, \phi^*)}{\partial \theta} + \frac{\mathcal{L}_{outer}(\theta, \phi^*)}{\partial \phi^*}\frac{\partial \phi^*}{\partial \theta} \tag{24}$$

where is the algorithm procedure to solve Eq1, i.e. SGD. While is just the gradient of the loss evaluated at the solution of the inner problem and can be computed efficiently with standard automatic backpropagation, requires further attention. Since the solution to $C_{\phi^*}$ is implemented via and iterative method (SGD), one strategy would be to compute this gradient would be to backpropagate trough the entire optimization trajectory in the inner loop. This strategy however is computational inefficient for many steps, and can suffer also from vanishing gradient problems.

# C  Experimental details

All experiments were performed on a single gpu NVIDIA RTX 3080Ti and implemented with the Pytorch library [69].

## C.1  Datasets

We evaluate our method on a synthetic setting on the following benchmarks: `DSprites`, `AbstractDSprites`[62], `3Dshapes` [14],`SmallNorb` [49], `Cars3D`[71] and the semi-synthetic `Waterbirds` [73].

For domain generalization and domain adaptation tasks, we evaluate our method on the [32] and [44] benchmarks, using the following datasets: `PACS`[52], `VLCS`[3], `OfficeHome`[87] `Camelyon17`[6], `CivilComments` [12].

**Dataset descriptions**

The `Waterbirds` dataset [73] is a synthetic dataset where images are composed of cropping out birds from photos in the `Caltech-UCSD Birds-200-2011 (CUB)` dataset [89] and transfer-

ring them onto backgrounds from the Places dataset [104]. The dataset contains a large percentage of training samples ($\approx \%95$) which are spuriously correlated with the background information.

The `CivilComments` is a dataset of textual reviews annotated with demographics information for the task of detecting toxic comments. Prior work has shown that toxicity classifiers can pick up on biases in the training data and spuriously associate toxicity with the mention of certain demographics [68, 20]. These types of spurious correlations can significantly degrade model performance on particular subpopulations [74].

The `PACS` dataset [52] is a collection of images coming from four different domains: *real images, art paintings, cartoon* and *sketch*. The `VLCS` dataset contains examples from 5 overlapping classes from the VOC2007 [23], LabelMe [72], Caltech-101 [24] , and SUN [98] datasets. The `OfficeHome` dataset contains 4 domains (Art, ClipArt, Product, real-world) where each domain consists of 65 categories.

The `Camelyon17` dataset, is a collection of medical tissue patches scanned from different hospital environments. The task is to predict whether a patch contain a benign or tumoral tissue. The different hospitals represent the different domains in this problem, and the aim is to learn a predictor which is robust to changes in factors of variation across different hospitals.

## C.2 Models

For synthetic datasets we use a CNN module for the backbone $g\theta$ following the architecture in Table 5. For real datasets that use images as modality we use a `ResNet50` architecure as backbone pretrained on the `Imagenet` dataset. For the experiments on the text modality we use `DistilBERT` model [76] with pretrained weights downloaded from HuggingFace [96].

## C.3 Synthetic experiments

Table 5: Convolutional architecture used in synthetic experiments.

| CNN backbone |
| --- |
| Input : $64 \times 64 \times$ number of channels |
| $4 \times 4$conv, 32 stride 2, padding 1, ReLU,BN |
| $4 \times 4$conv, 32 stride 2, padding 1, ReLU,BN |
| $4 \times 4$conv, 64 stride 2, padding 1, ReLU,BN |
| $4 \times 4$conv, 64 stride 2, padding 1, ReLU,BN |
| FC, 256, Tanh |
| FC, $d$ |

**Task generation**. For the synthetic experiments we have access to the ground truth factors of variations $\mathcal{Z}$ for each dataset. The task generation procedure relies on two hyperparameters: the first one is an index set $\mathbb{S}$ of possible factors of variations on which the distribution of tasks can depend on. The latter hyperparameter $K$, set the maximum number of factors of variations on which a single task can depend on. Then a task $t$ is sampled drawing a number $k_t$ from $\{1...K\}$, and then sampling randomly a subset $S$ of size $|\mathbb{S}| - k_t$ from $\mathbb{S}$. The resulting set $S$ will be the set indexing the factors of variation in Z on which the task $t$ is defined. In this setting restrict ourselves to binary task: for each factors in $S$, we sample a random value $v$ for it. The resulting set of values $V$, will determine uniquely the binary task.

Before selecting $v \in V$ we quantize the possible choices corresponding to factors of variations which may have more than six values to 2. We remark that this quantization affect only the task label definition. For examples for x axis factor, we consider the object to be on the left if its x coordinate is less than the medial axis of the image, on the right otherwise. The `DSprites` dataset has the following set of factors of variations $Z_{dsprites} = \{shape, size, angle, x_{pos}, y_{pos}\}$ and example of task is *There is a big object on the right* where $k_t = 2$ the affected factors are $size, x_{pos}$. Another example is *There is a small heart on the top left* , where $k_t = 4$ the affected factors are $shape, size, x_{pos}, y_{pos}$. Obervations are labelled positively of negatively if their corresponding factors of variations matching in the values with the one specified by the current task.

We then samples random query $Q$ and support $U$ set of samples balanced with respect to postive and negative labels of task task $t$, using stratified sampling.

## C.4 Experiments on domain shifts

For the domain generalization and few-shot transfer learning experiments we put ourselves in the same settings of [32, 44] to ensure a fair comparison. Namely, for each dataset we use the same augmentations, and same backbone models.

For solving the inner problem in Equation 5, we used Adam optimizer [42], with a learning rate of $1e-2$, momentum $0.99$, with the number of gradient steps varying from $50$ to $100$, in domain shifts experiments.

**Task generation**. The task (or episode) sampling procedure is done as follows: each task is a multiclass classification problem: we set the number of classes $C$ to $C = 5$ when the original number of classes $K_{train}$ in the dataset is higher than five, i.e. $K_{train} > 5$. Otherwise we set $C = K_{train}$. During training, the sizes of the support set $U$ and query sets $Q$ where set to $|U| = 25, |Q| = 15$ similar to as done in prior meta-learning literature [50, 18]. Changing these parameters has similar effects from what has been observed in many meta learning approaches(e.g. [50, 18]).

For binary datasets such as Camelyon17 or Waterbirds the possible classes to be predicted are always the same across tasks: what is changing is the composition of $U$ and $Q$. Keeping their cardinality low, we ensure that some tasks will not contain spurious correlation that may be present in the dataset, while other ones will still retain it, and the regularizers will satisfy solutions which discards the spurious information. We can observe evidence of this in the experimental results in Tables 3, 4 and qualitatively in Figure 8.

## C.5 Selection of $\alpha$ and $\beta$

To find the best regularization parameters $\alpha, \beta$ weighting the sparsity and feature sharing regularizers in Equation 1 respectively, we perform model selection according to the highest accuracy on a validation set. We report in Table 6 the value selected for each experiment.

Table 6: Selected values for $\alpha$ and $\beta$ for all experiments, applying model selection on validation set.

| Experiment | $\alpha$ | $\beta$ |
|---|---|---|
| Table 1 | 1e-2 | 0.15 |
| Table 2 | 1e-2 | 5e-2 |
| Table 3 | 2.5e-3 | 5e-2 |
| Table 4 | 1.5e-3 | 1e-2 |
| Table 5, 6 | 2.5e-3 | 1e-2 |
| Table 7 | 2.5e-3 | 1e-2 |

# D Additional results

## D.1 Synthetic experiments

**Enforcing disentanglement**: In Table 7 we report diverse disentanglement scores (DCI disentanglement, DCI completeness, DCI informativeness) on the `DSprites`, `3DShapes`, `SmallNorb`,`Cars` datasets, showing that the sparsity and feature sharing regularizers effectively enforce disentanglement.

**The role of minimality**. In Figure 7 we show the qualitative results accompanying Figure 3. The qualitative results in the Figure are produced visualizing matrices of feature importance [57] computed fitting Gradient Boosted Trees (GBT) on the learned representations w.r.t. task labels, and on the factors of variations w.r.t. task labels and compare the results. In each matrix the x axis represents the tasks and the y axis the features, and each entries the amount of feature importance (which goes from 0 to 1). In Figure 6 we show the same experiment on the `3DShapes` dataset.

Table 7: *Enforcing disentanglement*. DCI [22] disentanglement, completeness and informativeness scores and ID accuracy on test samples for a model trained without enforcing sufficiency and minimality (top row), and model with the regularizers activated (bottom row). While attaining similar performance on accuracy, the model with the activated regularizer always show higher disentanglement. See Table for additional scores.

|  | DSprites | 3DShapes | SmallNorb | Cars |
|---|---|---|---|---|
| *Without regularization* | | | | |
| DCI Disentanglement | 16.6 | 44.4 | 16.5 | 60.5 |
| DCI Completeness | 17.5 | 39.1 | 12.9 | 50.8 |
| DCI Informativeness | 88.0 | 87.6 | 90.5 | 95.5 |
| *With regularization* | | | | |
| DCI Disentanglement | 69.9 | 87.7 | 60.5 | 92.3 |
| DCI Completeness | 72.3 | 88.4 | 63.2 | 57.1 |
| DCI Informativeness | 96.0 | 95.7 | 95.4 | 99.7 |

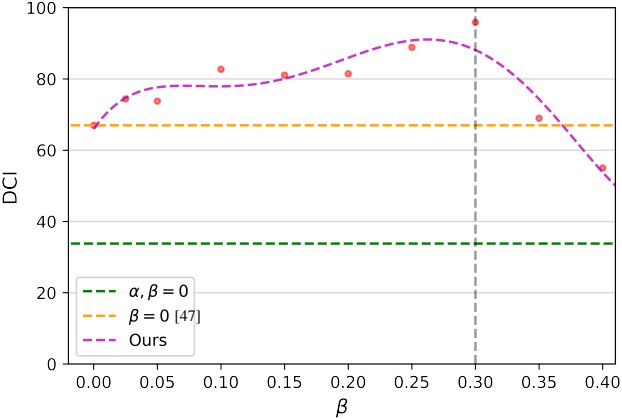

Figure 6: *Role of minimality (3DShapes)*: We plot the DCI disentanglement metric of a set of models (*red dots*) trained on fixed tasks from `3Dshapes`: Training without regularizers leads to no disentanglement (*green*). Enforcing sparsity alone (*yellow*, akin to [47]) achieves good disentanglement ($DCI = 67.0$), but some features may be split or duplicated. Enforcing both minimality and sparse sufficiency (*magenta*) attains the best $DCI$ (95.9). When $\beta$ is too high ($> 0.25$) activated features collapses into few clusters with respect to tasks.

**Task compositional generalization**. In Table 9 we show the quantitative results accompanying Figure 4.

### D.2 Properties of the learned representations

**Feature sufficiency.** The sufficiency property is crucial for robustness to spurious correlations in the data. If the model can learn and select the relevant features for a task, while ignoring the spurious ones, sufficiency is satisfied, resulting in robust performance under subpopulation shifts, as shown in Tables 10 and 4. To get qualitative evidence of the sufficiency in the representations, in Figure 8 we show the saliency maps computed from the activations of our model and a corresponding model trained with ERM. Our model can learn features specific to the subject of the image, which are

Table 8: Quantitative results accompanying Figure 7

|  | $\alpha = 0, \beta = 0$ | $\alpha = 1e-2, \beta = 0$ | $\alpha = 1e-2, \beta = 0.2$ | $\alpha = 1e-2, \beta = 0.4$ |
|---|---|---|---|---|
| DCI | 27.8 | 71.9 | 98.8 | 30.5 |

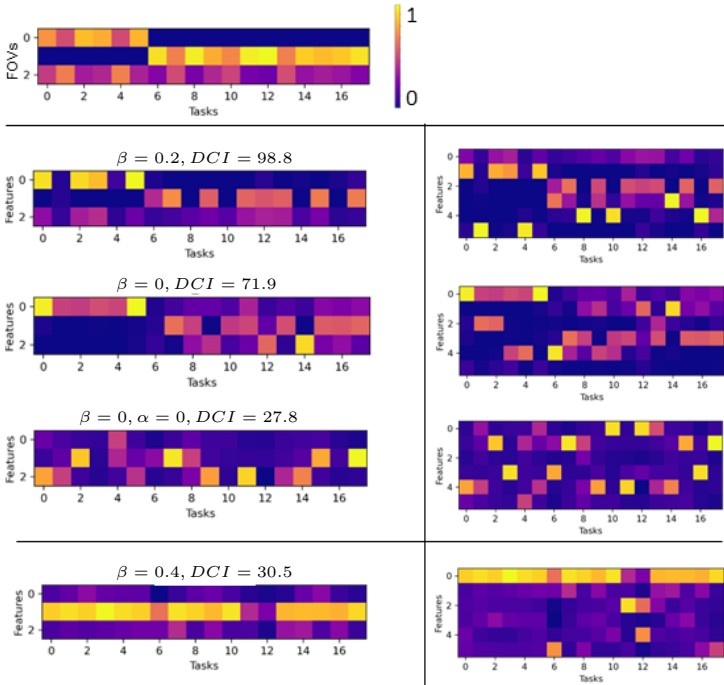

Figure 7: Qualitative dependency of disentanglement from the weight of our penalties ($\alpha = 0.01$ unless otherwise specified). The model that attains the best disentanglement ($DCI = 98.8$) uses both. *Left column, top*: ground-truth importance weights of each latent factor for each task. *Right column*: we train models with different $\beta$ and visualize the weights assigned to each learned feature on each task. *Left column*: to determine whether the model recover the ground-truth latents, we select the 3 top features and compare their assigned weights on different tasks with the ground-truth weights. *Bottom row*: example of a failure case with high $\beta$.

Table 9: *Task compositional generalization*: Mean accuracy over 100 random tasks reported for group of tasks of growing support (*second, third, fourth column*) for a model trained without inductive biases (*top row*) and enforcing them (*bottom row*). The latter show better compositional generalization resulting from the properties enforced on the representation

|  | Acc ID | DCI | $|S| = 3$ | $|S| = 4$ | $|S| = 5$ |
|---|---|---|---|---|---|
| *No reg* | 88.7 | 22.8 | 72.6 | 63.3 | 59.9 |
| $\alpha, \beta$ | **93.2** | **59.4** | **83.0** | **78.8** | **76.8** |

relevant for classification, while ignoring background information. This can be observed in both correctly classified (bottom row) and misclassified (top row) samples by ERM. In contrast, ERM activates features in the background and relies on them for prediction.

**Feature sharing.** In this section, we study the minimality properties of the representations learned by our method. To achieve this, we conduct the following experiment. We randomly draw 14 tasks from the $\sum_{i=1}^{3} \binom{4}{i}$ possible combinations of the four domains in the `PACS` dataset. We use the data from these tasks to fit the linear head and test the model accuracy on the OOD domain (e.g. the *sketch* domain). In Figure 9, we show the performance on each task, ordered on the x axis according to OOD accuracy of a model trained with ERM (in yellow). We also report the fraction of activated features (in blue) shared between each task and the OOD task, and the same(red) for the ERM model. The fraction of activated features is computed by looking at the matrix of coefficients of the sparse linear head $\phi \in \mathbb{R}^{M \times C}$, where $M$ is the number of features and $C$ the number of classes, after fitting on each task. Specifically, is computed as $\frac{\sum_m [\tilde{\phi}_\epsilon \cap \tilde{\phi}_\epsilon^{OOD}]}{\sum_m [\tilde{\phi}_\epsilon \cup \tilde{\phi}_\epsilon^{OOD}]}$ where $\tilde{\phi}_\epsilon = \frac{1}{C} \sum_c |\phi_{m,c}| > \epsilon$ and $\phi^{OOD}$

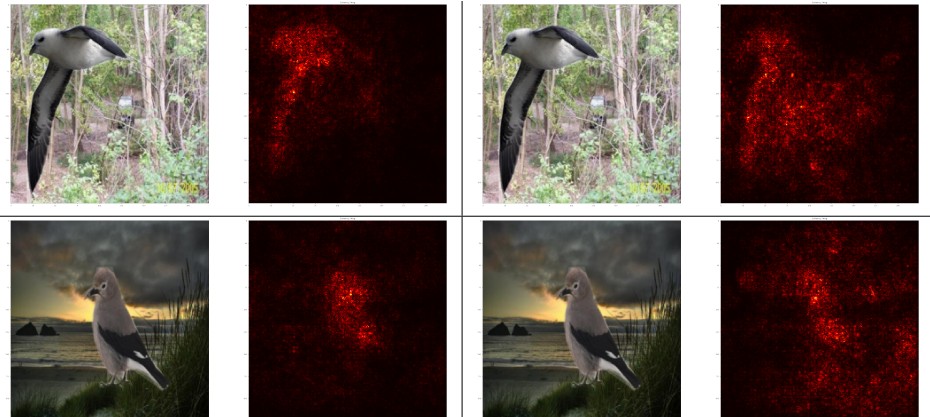

Figure 8: *Feature sufficiency: Left*, pairs of random samples and saliency maps computed on activations with our method. All samples are correctly classified. *Right*, corresponding saliency maps [1] an ERM based method: the first row is misclassifed by the network, the last is correctly classified. The ERM model depends on features from the background, resulting in a higher prediction error on mixed subdomains. Our model is robust to spurious correlations and satisfies the sufficiency assumptions.

is the matrix of coefficient of the OOD task. We set $\epsilon = 0.01$. From Figures 9 and 10 we draw the following conclusions: (i) When the accuracy of the ERM decreases (i.e., the current task is farther from the OOD test task), our method is still able to retain a high and consistent accuracy, demonstrating that our features are more robust out-of-distribution. This is further supported by the higher number of shared features compared to ERM, as we move away from the testing domain. (ii) The correlation between the fraction of shared features and the accuracy OOD demonstrates that the method is able to learn general features that transfer well to unseen domains, thanks to the minimality constraint. Additionally, this measure serves as a reliable indicator of task distance, as discussed in the next section. (iii) Even though the same sparse linear head is used on top of the ERM and our features, our method is able to achieve better OOD performance with fewer features, further demonstrating our feature minimality.

### D.3 CivilComments

See Table 10 for the quantitative results accompanying to Figure 5 in the paper and 11 for result on groups on the civil comments dataset.

Table 10: *Quantitative results on CivilComments*: we report the accuracy on test averaged across all demographic groups (*left*), and the worst group accuracy (*right*). We show that our method performs similarly in terms of average accuracy and outperforms in terms of worst group accuracy, without using any knowledge on the group composition in the training data. This Table accompanies Figure 5

|      | avg acc          | worst group acc     |
| ---- | ---------------- | ------------------- |
| ERM  | **92.2**         | 56.5                |
| DRO  | 90.2             | 69                  |
| Ours | $91.2 \pm 0.2$   | **75.45**$\pm 0.1$  |

### D.4 Full results Domain generalization

We report here comparison with several methods in the domain generalization literature, namely [99, 10, 51, 53, 26, 54, 65, 102, 36, 45].

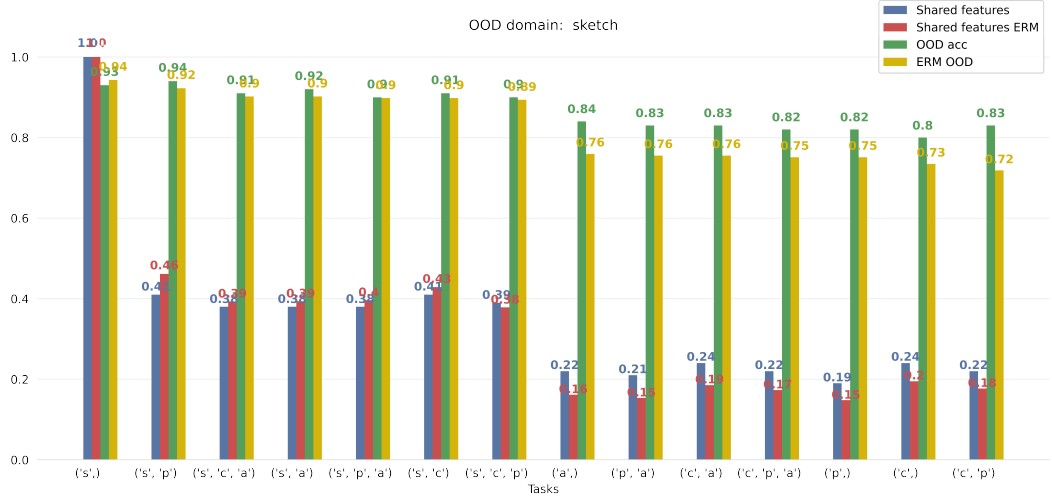

Figure 9: *Fraction of shared features VS accuracy*. Barplot of OOD accuracies on the *Sketch* domain for our model (green) and ERM (yellow) on the 14 tasks sampled from PACS, along with the fraction of shared features with the OOD domain for each task (blue for our model, red for ERM). Each task is sampled from a single domain or from the intersections of domains. Tasks are labelled according to the sampling domain on the x axis. The fraction of shared features and OOD accuracy have a correlation coefficient of 97.5.

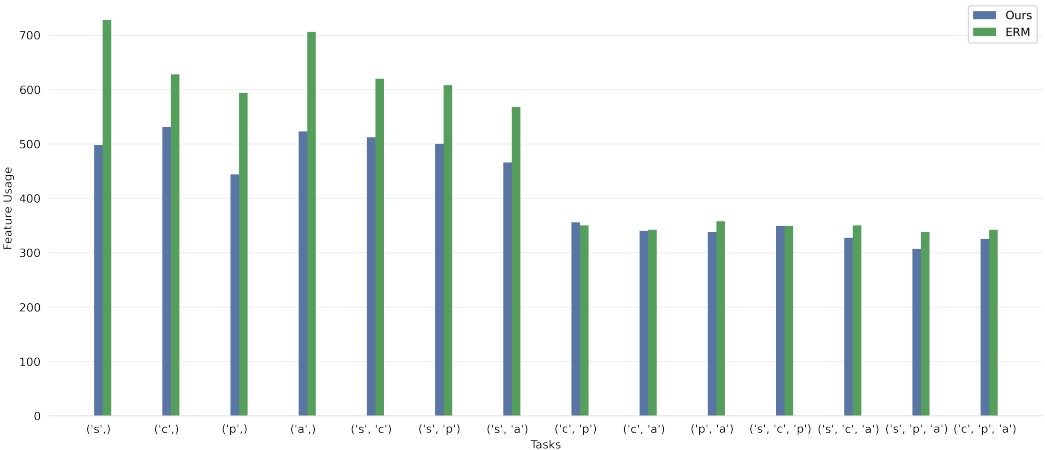

Figure 10: Barplot of feature usage (number of activated features) for each task for our model (blue) and ERM model (green) referring to the experiment in Figure 9. Our method uses fewer features than ERM while also generalizing better.

Table 11: Civilcomments quantitative results pergroup.

|  | Male | Female | LGBTQ | Christian | Muslim | Other religion | Black | White |
|---|---|---|---|---|---|---|---|---|
| *GroupDRO* | | | | | | | | |
| Toxic | $75.1 \pm 2.1$ | $73.7 \pm 1.5$ | $73.7 \pm 4$ | $69.2 \pm 2.0$ | $72.1 \pm 2.6$ | $72.0 \pm 2.5$ | $79.6 \pm 2.2$ | $78.8 \pm 1.7$ |
| Non Toxic | $88.4 \pm 0.7$ | $90.0 \pm 0.6$ | $76.0 \pm 3.6$ | $92.6 \pm 0.6$ | $80.7 \pm 1.9$ | $87.4 \pm 0.9$ | $72.2 \pm 2.3$ | $73.4 \pm 1.4$ |
| *Ours* | | | | | | | | |
| Toxic | $87.94 \pm 0.07$ | $89.17 \pm 0.05$ | $77.25 \pm 0.16$ | $92.25 \pm 0.16$ | $80.6 \pm 0.29$ | $87.79 \pm 0.26$ | $75.45 \pm 0.17$ | $78.35 \pm 0.02$ |
| Non toxic | $91.62 \pm 0.11$ | $91.52 \pm 0.11$ | $91.71 \pm 0.16$ | $91.11 \pm 0.1$ | $91.81 \pm 0.12$ | $91.32 \pm 0.1$ | $90.82 \pm 0.12$ | $92.04 \pm 0.11$ |

### D.4.1 VLCS

| Algorithm | C | L | S | V | Avg |
|---|---|---|---|---|---|
| ERM | 97.7 ± 0.4 | 64.3 ± 0.9 | 73.4 ± 0.5 | 74.6 ± 1.3 | 77.5 |
| IRM | 98.6 ± 0.1 | 64.9 ± 0.9 | 73.4 ± 0.6 | 77.3 ± 0.9 | 78.5 |
| GroupDRO | 97.3 ± 0.3 | 63.4 ± 0.9 | 69.5 ± 0.8 | 76.7 ± 0.7 | 76.7 |
| Mixup | 98.3 ± 0.6 | 64.8 ± 1.0 | 72.1 ± 0.5 | 74.3 ± 0.8 | 77.4 |
| MLDG | 97.4 ± 0.2 | 65.2 ± 0.7 | 71.0 ± 1.4 | 75.3 ± 1.0 | 77.2 |
| CORAL | 98.3 ± 0.1 | **66.1** ± 1.2 | 73.4 ± 0.3 | 77.5 ± 1.2 | **78.8** |
| MMD | 97.7 ± 0.1 | 64.0 ± 1.1 | 72.8 ± 0.2 | 75.3 ± 3.3 | 77.5 |
| DANN | 99.0 ± 0.3 | 65.1 ± 1.4 | 73.1 ± 0.3 | 77.2 ± 0.6 | 78.6 |
| CDANN | 97.1 ± 0.3 | 65.1 ± 1.2 | 70.7 ± 0.8 | 77.1 ± 1.5 | 77.5 |
| MTL | 97.8 ± 0.4 | 64.3 ± 0.3 | 71.5 ± 0.7 | 75.3 ± 1.7 | 77.2 |
| SagNet | 97.9 ± 0.4 | 64.5 ± 0.5 | 71.4 ± 1.3 | 77.5 ± 0.5 | 77.8 |
| ARM | 98.7 ± 0.2 | 63.6 ± 0.7 | 71.3 ± 1.2 | 76.7 ± 0.6 | 77.6 |
| VREx | 98.4 ± 0.3 | 64.4 ± 1.4 | 74.1 ± 0.4 | 76.2 ± 1.3 | 78.3 |
| RSC | 97.9 ± 0.1 | 62.5 ± 0.7 | 72.3 ± 1.2 | 75.6 ± 0.8 | 77.1 |
| **Ours** | **98.1** ± 0.2 | 63.4 ± 0.5 | **73.9** ± 0.8 | **78.2** ± 0.7 | 78.4 |

### D.4.2 PACS

| Algorithm | A | C | P | S | Avg |
|---|---|---|---|---|---|
| ERM | 84.7 ± 0.4 | 80.8 ± 0.6 | 97.2 ± 0.3 | 79.3 ± 1.0 | 85.5 |
| IRM | 84.8 ± 1.3 | 76.4 ± 1.1 | 96.7 ± 0.6 | 76.1 ± 1.0 | 83.5 |
| GroupDRO | 83.5 ± 0.9 | 79.1 ± 0.6 | 96.7 ± 0.3 | 78.3 ± 2.0 | 84.4 |
| Mixup | 86.1 ± 0.5 | 78.9 ± 0.8 | 97.6 ± 0.1 | 75.8 ± 1.8 | 84.6 |
| MLDG | 85.5 ± 1.4 | 80.1 ± 1.7 | 97.4 ± 0.3 | 76.6 ± 1.1 | 84.9 |
| CORAL | 88.3 ± 0.2 | 80.0 ± 0.5 | 97.5 ± 0.3 | 78.8 ± 1.3 | 86.2 |
| MMD | 86.1 ± 1.4 | 79.4 ± 0.9 | 96.6 ± 0.2 | 76.5 ± 0.5 | 84.6 |
| DANN | 86.4 ± 0.8 | 77.4 ± 0.8 | 97.3 ± 0.4 | 73.5 ± 2.3 | 83.6 |
| CDANN | 84.6 ± 1.8 | 75.5 ± 0.9 | 96.8 ± 0.3 | 73.5 ± 0.6 | 82.6 |
| MTL | 87.5 ± 0.8 | 77.1 ± 0.5 | 96.4 ± 0.8 | 77.3 ± 1.8 | 84.6 |
| SagNet | 87.4 ± 1.0 | 80.7 ± 0.6 | 97.1 ± 0.1 | 80.0 ± 0.4 | 86.3 |
| ARM | 86.8 ± 0.6 | 76.8 ± 0.5 | 97.4 ± 0.3 | 79.3 ± 1.2 | 85.1 |
| VREx | 86.0 ± 1.6 | 79.1 ± 0.6 | 96.9 ± 0.5 | 77.7 ± 1.7 | 84.9 |
| RSC | 85.4 ± 0.8 | 79.7 ± 1.8 | 97.6 ± 0.3 | 78.2 ± 1.2 | 85.2 |
| **Ours** | **86.7** ± 0.1 | **83.5** ± 0.8 | **97.8** ± 0.1 | **83.1** ± 0.1 | **87.5** |

### D.4.3 OfficeHome

| Algorithm | A | C | P | R | Avg |
|---|---|---|---|---|---|
| ERM | 61.3 ± 0.7 | 52.4 ± 0.3 | 75.8 ± 0.1 | 76.6 ± 0.3 | 66.5 |
| IRM | 58.9 ± 2.3 | 52.2 ± 1.6 | 72.1 ± 2.9 | 74.0 ± 2.5 | 64.3 |
| GroupDRO | 60.4 ± 0.7 | 52.7 ± 1.0 | 75.0 ± 0.7 | 76.0 ± 0.7 | 66.0 |
| Mixup | 62.4 ± 0.8 | 54.8 ± 0.6 | 76.9 ± 0.3 | 78.3 ± 0.2 | 68.1 |
| MLDG | 61.5 ± 0.9 | 53.2 ± 0.6 | 75.0 ± 1.2 | 77.5 ± 0.4 | 66.8 |
| CORAL | 65.3 ± 0.4 | 54.4 ± 0.5 | 76.5 ± 0.1 | 78.4 ± 0.5 | 68.7 |
| MMD | 60.4 ± 0.2 | 53.3 ± 0.3 | 74.3 ± 0.1 | 77.4 ± 0.6 | 66.3 |
| DANN | 59.9 ± 1.3 | 53.0 ± 0.3 | 73.6 ± 0.7 | 76.9 ± 0.5 | 65.9 |
| CDANN | 61.5 ± 1.4 | 50.4 ± 2.4 | 74.4 ± 0.9 | 76.6 ± 0.8 | 65.8 |
| MTL | 61.5 ± 0.7 | 52.4 ± 0.6 | 74.9 ± 0.4 | 76.8 ± 0.4 | 66.4 |
| SagNet | 63.4 ± 0.2 | 54.8 ± 0.4 | 75.8 ± 0.4 | 78.3 ± 0.3 | 68.1 |
| ARM | 58.9 ± 0.8 | 51.0 ± 0.5 | 74.1 ± 0.1 | 75.2 ± 0.3 | 64.8 |
| VREx | 60.7 ± 0.9 | 53.0 ± 0.9 | 75.3 ± 0.1 | 76.6 ± 0.5 | 66.4 |
| RSC | 60.7 ± 1.4 | 51.4 ± 0.3 | 74.8 ± 1.1 | 75.1 ± 1.3 | 65.5 |
| **Ours** | **66.7** ± 0.1 | **56.3** ± 0.7 | **79.2** ± 0.5 | **81.3** ± 0.4 | **70.9** |

## D.5 Few-shot transfer learning

Results on few-shot transfer learning on datasets `PACS,VLCS,OfficeHome,Waterbirds` in Tables 12,13,14 and 15.

Table 12: Results few-shot transfer learning on PACS

| Dataset/Algorithm | OOD accuracy (by domain) | | | | |
|---|---|---|---|---|---|
| **PACS 1-shot** | S | A | P | C | Average |
| ERM | $72.3 \pm 0.3$ | $80.4 \pm 0.09$ | $93.3 \pm 4.1$ | $75.8 \pm 2.6$ | 80.5 |
| Ours | $\mathbf{75.4 \pm 3}$ | $\mathbf{81.7 \pm 0.8}$ | $\mathbf{98.0 \pm 0.8}$ | $\mathbf{71 \pm 5.2}$ | **81.5** |
| **PACS 5-shot** | S | P | A | C | Average |
| ERM | $84.9 \pm 1.1$ | $85.7 \pm 0.08$ | $98.6 \pm 0.0$ | $79.1 \pm 0.9$ | 87.1 |
| Ours | $\mathbf{85.0 \pm 0.1}$ | $\mathbf{86.7 \pm 0.8}$ | $97.8 \pm 0.1$ | $\mathbf{83.5 \pm 0.1}$ | **88.3** |
| **PACS 10-shot** | S | P | A | C | Average |
| ERM | $81.0 \pm 0.1$ | $88.9 \pm 0.1$ | $97.4 \pm 0.0$ | $84.2 \pm 0.9$ | 87.9 |
| Ours | $\mathbf{86.2 \pm 0.5}$ | $\mathbf{90.0 \pm 0.8}$ | $\mathbf{98.9 \pm 0.1}$ | $\mathbf{86.6 \pm 0.1}$ | **90.4** |

Table 13: results few-shot transfer learning on VLCS

| Dataset/Algorithm | OOD accuracy (by domain) | | | | |
|---|---|---|---|---|---|
| **VLCS 1-shot** | C | L | V | S | Average |
| ERM | $98.9 \pm 0.4$ | $32.7 \pm 16.2$ | $59.8 \pm 10.7$ | $47.5 \pm 11.2$ | 59.7 |
| Ours | $98.6 \pm 0.3$ | $\mathbf{51.0 \pm 4.9}$ | $\mathbf{61.2 \pm 9.8}$ | $\mathbf{61.9 \pm 9.7}$ | **68.2** |
| **VLCS 5-shot** | C | L | V | S | Average |
| ERM | $99.4 \pm 0.2$ | $50.0 \pm 6.2$ | $71.9 \pm 3.2$ | $65.3 \pm 2.8$ | 71.7 |
| Ours | $98.9 \pm 0.4$ | $\mathbf{56.0 \pm 6.2}$ | $\mathbf{73.4 \pm 1.4}$ | $\mathbf{69.8 \pm 2.0}$ | **74.5** |
| **VLCS 10-shot** | C | L | V | S | Average |
| ERM | $99.5 \pm 0.2$ | $52.6 \pm 5.0$ | $74.8 \pm 3.8$ | $69.1 \pm 2.4$ | 74.0 |
| Ours | $99.1 \pm 0.2$ | $\mathbf{65.0 \pm 6.2}$ | $74.4 \pm 1.9$ | $\mathbf{70.8 \pm 2.3}$ | **77.3** |

Table 14: results few-shot transfer learning on OfficeHome

| Dataset/Algorithm | OOD accuracy (by domain) | | | | |
|---|---|---|---|---|---|
| **OfficeHome 1-shot** | C | A | P | R | Average |
| ERM | $40.2 \pm 2.4$ | $52.7 \pm 2.6$ | $68.1 \pm 1.7$ | $64.6 \pm 1.8$ | 56.4 |
| Ours | $\mathbf{41.4 \pm 1.7}$ | $\mathbf{54.5 \pm 2.0}$ | $\mathbf{68.5 \pm 2.7}$ | $\mathbf{69.0 \pm 1.5}$ | **58.4** |
| **OfficeHome 5-shot** | C | A | P | R | Average |
| ERM | $63.2 \pm 0.4$ | $73.3 \pm 0.8$ | $84.1 \pm 0.4$ | $82.0 \pm 0.8$ | 75.7 |
| Ours | $\mathbf{66.2 \pm 1.2}$ | $\mathbf{75.1 \pm 1.0}$ | $83.6 \pm 0.5$ | $\mathbf{83.1 \pm 0.8}$ | **77.0** |
| **OfficeHome 10-shot** | C | A | P | R | Average |
| ERM | $71.1 \pm 0.4$ | $80.5 \pm 0.5$ | $87.5 \pm 0.3$ | $84.9 \pm 0.5$ | 81.0 |
| Ours | $\mathbf{72.2 \pm 1.2}$ | $\mathbf{81.8 \pm 0.5}$ | $\mathbf{87.5 \pm 0.2}$ | $\mathbf{86.3 \pm 0.4}$ | **82.0** |

## D.6 Feature sharing on `PACS`

See Figure 11 for additional results on all domains in `PACS`.

## D.7 Task similarity

We show that our method enables direct extraction of a task representation and a metric for task similarity from our model and its feature space. We propose to use the coefficients of the fitted linear

Table 15: results few-shot transfer learning Waterbirds

| Dataset/Algorithm | OOD accuracy (by domain) | | | | |
|---|---|---|---|---|---|
| **Waterbirds 1-shot** | LL | LW | WL | WW | Average |
| ERM | $99.1 \pm 1.1$ | $43.8 \pm 16.5$ | $79.5 \pm 10.2$ | $86.7 \pm 8.2$ | 79.8 |
| Ours | $\mathbf{95.2} \pm 8.1$ | $\mathbf{81.9} \pm 9.5$ | $\mathbf{80.7} \pm 5.5$ | $\mathbf{95.9} \pm 1.2$ | **88.4** |
| **Waterbirds 5-shot** | LL | LW | WL | WW | Average |
| ERM | $96.3 \pm 5.0$ | $58.7 \pm 17.2$ | $80.1 \pm 12.6$ | $84.1 \pm 12.7$ | 79.8 |
| Ours | $\mathbf{98.8} \pm 1.8$ | $\mathbf{75.4} \pm 9.0$ | $\mathbf{81.6} \pm 14.0$ | $\mathbf{94.8} \pm 1.8$ | **87.6** |
| **Waterbirds 10-shot** | LL | LW | WL | WW | Average |
| ERM | $94.2 \pm 4.2$ | $73.0 \pm 11.6$ | $80.4 \pm 6.3$ | $89.3 \pm 3.3$ | 84.2 |
| Ours | $\mathbf{98.2} \pm 0.9$ | $\mathbf{82.6} \pm 5.9$ | $\mathbf{80.7} \pm 6.3$ | $\mathbf{95.5} \pm 1.4$ | **89.2** |

heads $f_{\phi_t^*}$ on a given task as a *representation for that task*. Specifically we transform the optimal coefficients $\phi^*$ in a $M$-dimensional vector space (here $M$ is the number of features) by simply computing $\sum_c |\phi_{t,m,c}^*|$, and discretize them by a threshold $\epsilon$. The resulting binary vectors, together with a distance metric (we choose the Hamming distance), form a discrete metric space of tasks. We preliminary verify how the proposed representation and metric behave on `MiniImagenet` [88] below.

We sample 160 tasks from 10 groups from , where each group has the same class support, i.e. $t_1, t_2 \in G_i \mapsto Supp(t_1) == Supp(t_2) \forall i$. We then fit the linear heads independently on each task (i.e. not using the feature sharing regularizer). Then we compute the discrete task representation and project the resulting vector space in a two dimensional vector space using tSNE [92]. The clusters obtained in this space correspond exactly to the group identities (visualized in color in Figure 12).

### D.8    Comparison with metalearning baselines

In Table 16, we further compare our method on meta learning benchmarks, namely `Mini Imagenet` [88] and CIFAR-FS [9] with different approaches in the literature based on meta learning [81, 66, 18, 47].

In Figure 13 we compare the predicting performance of our method and capacity to leverage shared knowledge between task, comparing with backbone trained with protopical network approach. We sample a set of task with different overlap, where the overlap between two task $t_1, t_2$ is defined as $sim(t_1, t_2) = \frac{Supp(t_1) \cap Supp(t_2)}{Supp(t_1) \cup Supp(t_2)}$ indicating with $Supp(t_i)$ the support over classes in task $t_i$. We show that other than reaching a much higher accuracy the features of our model are able to be clustered at test time enabling to reach better performance on unseen task. As a matter of fact we can use the feature sharing regularizer at test time showing that there is a increasing trend in the performance, while the prototypical networks features just decreases being unable to share information across tasks at test time.

Table 16: Meta learning baselines, including concurrent work [47] which we significantly outperform.

| | Architecture | Cifar-FS (1 shot) | Cifar-FS ( 5 shot) | MiniImagenet(1 shot) | MiniImagenet (5 shot) |
|---|---|---|---|---|---|
| MAML | Conv32(x4) | - | - | $48.7 \pm 1.84$ | $63.11 \pm 0.66$ |
| Prototypical Net | Conv64(x4) | - | - | $49.42 \pm 0.78$ | $68.20 \pm 0.66$ |
| TADAM | ResNet12 | - | - | $58.5 \pm 0.56$ | $76.7 \pm 0.3$ |
| MetaOptNet | ResNet12 | $72.0 \pm 0.7$ | $84.2 \pm 0.5$ | $\mathbf{62.64} \pm 0.61$ | $\mathbf{78.63} \pm 0.46$ |
| MetaBaseline | WRN 28-10 | $\mathbf{76.58} \pm 0.68$ | $85.79 \pm 0.5$ | $59.62 \pm 0.66$ | $78.17 \pm 0.49$ |
| *Lachapelle et al*[47] | ResNet12 | - | - | $54.22 \pm 0.6$ | $70.01 \pm 0.51$ |
| Ours* | ResNet12 | $75.1 \pm 0.4$ | $\mathbf{86.9} \pm 0.19$ | $60.1 \pm 2$ | $76.6 \pm 0.1$ |

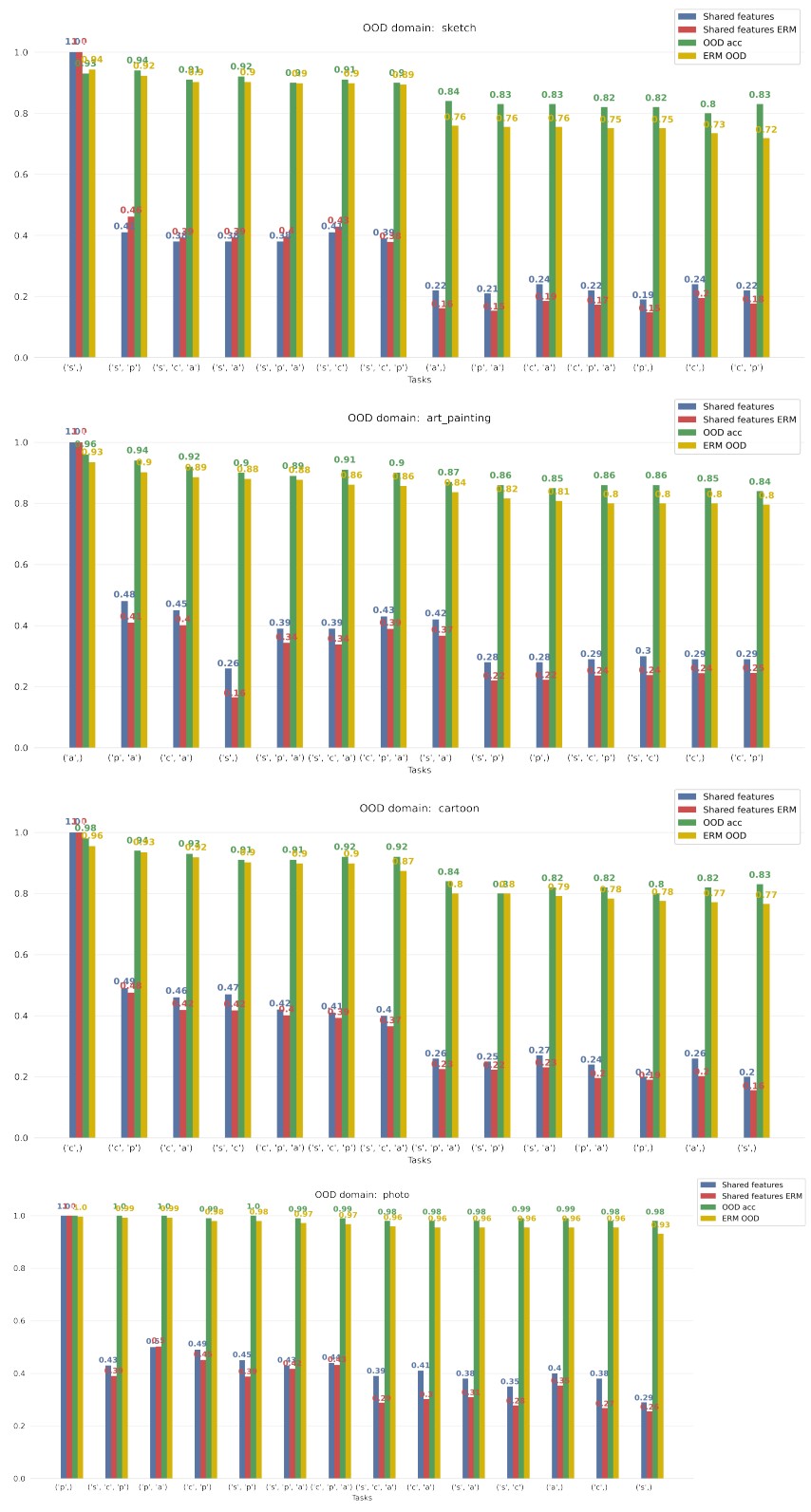

Figure 11: Additional results for all domains in PACS, separated by domain. The overall message of Figure 9 appear consistent across all domains.

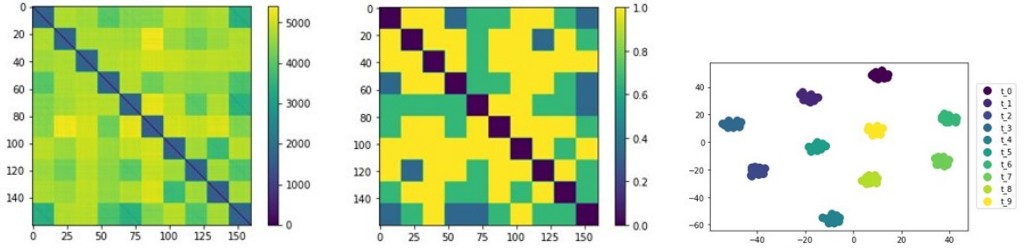

Figure 12: *Task Similarity.* We visualize the tSNE of the discrete task representation and observe that the clusters in this space corresponds to group identities.

### D.9 Sharing features at test time

Features can be enforced to be shared also at test time, simply by setting $\beta > 0$ to fit the linear head on top of the learned feature space. We observe the benefits of utilizing the feature sharing penalty at test time on the `Camelyon17` dataset in the fourth row of Table 17.

As highlighted in the main paper, retaining features which are shared across the training domains and cutting the ones that are domain-specific enable to perform better at test time, at the expenses of lower performance near the training distribution.

We analyzed in more depth this phenomenon in Figure 13. For this experiment we trained our model and a Prototypical network [81] one on the `MiniImagenet` dataset. Then we sampled 5 groups of tasks according to an average overlap measure between tasks. Between two task $t_1, t_2$ the overlap is defined as $sim(t_1, t_2) = \frac{Supp(t_1) \cap Supp(t_2)}{Supp(t_1) \cup Supp(t_2)}$. each group is made of 10 task. We then plot the performance at test time increasing the regularization parameter $\beta$, weighting the feature sharing. The outcome of the experiment is twofold: (i) we observe an increase in performance at test time, especially when tasks shows maximal overlap (i.e. they share more features) (ii) this is not the case with the pretrained backbone of [81] which shows almost monotonical decrease in the performance, i.e. enforcing the minimality property during training enables to use it as well at test time.

Further analysis on different datasets, and also on tuning strategies on the regularization parameter are promising directions for future work, to better understand when and how enforcing feature sharing is beneficial at test time.

Table 17: Camelyon17 quantitative results: we report accuracy both on ID and OOD splits. We show (last row) that feature sharing at test time, leads to more robust features on OOD test data.

|  | Validation(ID) | Validation (OOD) | Test (OOD) |
| --- | --- | --- | --- |
| ERM | 93.2 | 84 | 70.3 |
| CORAL | 95.4 | 86.2 | 59.5 |
| IRM | 91.6 | 86.2 | 64.2 |
| Ours | **93.2**±0.3 | **89.9**±0.6 | 74.1±0.2 |
| Ours($\beta > 0$ test) | 90.4±0.2 | 84.01±0.9 | **85.5**±0.6 |

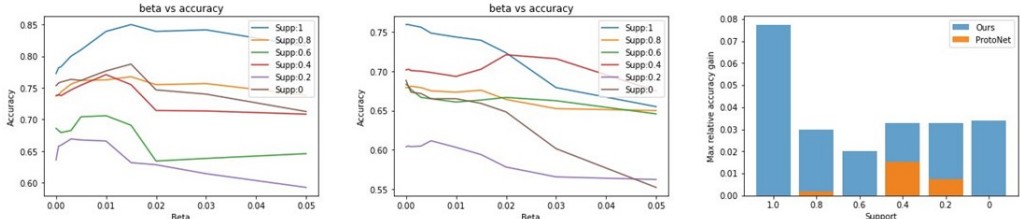

Figure 13: Enforcing feature sharing at test time. Our approach (on the left) is able to benefit from the feature sharing constraint at test time, while using the prototypical network backbone performance monotonically decrease (center). On the right we show the maximal performance gain for each group of tasks for the two approaches.