# OpenReview forum: "Leveraging sparse and shared feature activations for disentangled representation learning"
_NeurIPS.cc/2023/Conference — NeurIPS 2023 spotlight_

### Official Review · Reviewer_S5sy · 2023-06-30

**Soundness:** 3 good
**Presentation:** 2 fair
**Contribution:** 3 good
**Rating:** 7
**Confidence:** 4

**Summary:**

The paper proposes a novel regularization technique for disentangled representation learning. This regularizer is based on two principles formalized in the paper: 1) sparsity and 2) minimality. It assumes access to multiple tasks during training where each task uses only a subset of features, and different tasks share some of the features. The paper also provides theoretical results showing that these two principles are enough for the identification of disentangled factors. The proposed approach is validated, showing the significance of both regularization terms. Empirical results show the benefits of the proposed feature learning technique over standard ERM training.


**Strengths:**

-  The proposed two formalized principles (sparse sufficiency and minimality) and their sufficiency for disentangled representation learning are valuable contributions to the field.
-  The empirical study assesses the quality of the learned representations beyond disentanglement, showing the practical benefits of the proposed approach.
-  The overall idea and motivation are sound.


**Weaknesses:**

1. The paper lacks clarity; certain details and symbols' introductions are missing for better understanding (more in the question section). The writing can be improved.
    - For example, proposition 2.1 does not clearly state what S' and i are. Also, the symbol $f_t^*$ is not introduced. The proposition also misses subsequent discussion for better understanding.
    - The text has a number of mistakes. For example, L50: "a**n** entropy", L57: "co~~h~~exist", L77: "give**n**", L88: the formula has an extra parenthesis, L110: subscript $t$ is missing for $y$s.
2. The empirical evaluation is insufficient and lacks comparisons with other disentangled representation learning methods as well as few-shot/meta-learning methods.
    - Tab. 1 lacks other repr. learning methods.
    - Tab. 2 and 4 lack other few-shot/meta-learning methods. The main paper refers to Appendix for the latter mentioning a 10% improvement on MiniImageNet. However, Tab.15 in Appendix does not support the claim, with MetaOptNet being the best.
    - Fig. 5 lacks other domain adaptation methods.
3. It is not entirely clear how the sharing regularizer in (3) corresponds to the minimality principle introduced in Sec. 2.3. Also, it is unclear how exactly it favors clustering. Further clarifications are needed. (see questions for a more concrete question)


**Questions:**

1. How the DCI metric is defined exactly? The paper refers to [22], which has three separate metrics. Is DCI a combination of those? If yes, how are they combined?
2. The Waterbirds dataset has only two classes; how is task distribution defined?
3. Is regularization used or not when learning a novel test task? If yes, where do multiple tasks need for it come from?
4. For the experiments on DomainBed, it is stated that pretrained RN50 is used. How the proposed method influences the features learned in this case? Do you finetune it?
5. In the definition of the two principles, what is the difference between $f_t^*/f^*/f^*_{S_t}$? They are not properly introduced.
6. See 3 from weaknesses.
    - How does the sharing term "forces features to cluster across tasks and therefore be reused by different tasks, when useful" (L92-93)? For example, consider 3 FoVs $f_i$ and 3 binary classification tasks $t_i$ (with logits for simplification, i.e., M=1) corresponding to each combination of only 2 features with weights +1/-1, i.e., $y = \sigma(f_{S_t[1]} - f_{S_t[2]})$, $S_t \in \{(1, 2), (1, 3), (2, 3)\}$ In this case, $\tilde{\phi}_m = 2/6 = 1/3$ (removing the extra $1/TC$ term, see minor comments), resulting in a uniform distribution that results in high entropy (maximum possible). On the other hand, the sparsity and minimality are satisfied. It seems that the distribution should be over $t,c$ instead of $m$.
    - How does it correspond to the minimality principle introduced in Sec 2.3? The latter seems to suggest that no (smaller? see 9 below) subset of features can solve the task. Why having an L1 loss is not enough to satisfy this? If using fewer features is possible, the corresponding weights will be set to 0 following the L1 regularizer.
7. While the overall idea is sound, it is unclear why the meta-learning framework is necessary. Can the same technique be applied to the standard multi-task setting where different linear heads are learned simultaneously with a shared feature extractor?


Minor:
8. Why scaling by $1/TC$ in L88? $\tilde{\phi}_m$ is not a distribution in this case.
9. in L129 do you mean $S' \subset S_t$? Otherwise can include $S_t$ and be sufficient for classification.


**Limitations:**

Limitations are well-addressed.

---

> ### Author Rebuttal · Authors · 2023-08-10
>
> We thank the reviewer for all their suggestions, questions and comments. We will address them in the following. We remain available  during the discussion period for any further question or concern.
>
> ### **Clarity**
>
> We thank the reviewer for pointing out unclear sections and typos.
> Concerning Proposition 2.1, $S,S'$ correspond to two arbitrary instances of the index set $\mathcal{S}$, representing the support of two arbitrary tasks, and ${i}$ is a singleton. The assumption $p(S \cap S' = \{i\})> 0$ or  $p( \{i\} \in (S \cup S') -(S'\cap S))>0$  in Proposition 2.1 states that:
> - The probability of two tasks having a singleton intersection of support on the factor of variations is non zero
> - The probability that their difference of supports is a singleton is non zero.
>
> Intuitively, it means that there are two tasks that have either one factor of variation in common or have several factors in common but only one different. We will add a visualization example in the paper to help intuition.
> Finally, $f_t^*$ is defined as the unknown mixing function that defines uniquely a task $t$. $f_t^*$ and $f_{S_t}^*$ corresponds to the same object, thank you for spotting this and apologize for the confusion.
> We will incorporate these considerations in the theoretical section, to make it clearer and more intuitive. Nevertheless, we respectfully remark that clarity was a strength pointed out by R1 and R2.
>
>
>
> ### **Empirical evaluation**
>
>
> - **Synthetic**: The experiments in section 4.1 (Table 1) serve to validate our theory and demonstrate that sufficiency and minimality lead to disentangled representations. It would be non trivial to compare to other methods which are not in the multi-task learning setting (e.g. $\beta$-VAE), as their assumptions are different. Nevertheless, we compare with [47] in Figure 3, which shares a similar setting, showing that sparsity alone (yellow dashed line) cannot achieve the best DCI score when the number of factors of variation is unknown. **We included  a similar experiment to further compare with [47] on the 3DShapes dataset, in Figure A, in the attached PDF.**
>
> - **Domain shift** : For the experiments on domain shifts we compare with other 13 baselines in Appendix D.4. We reported ERM in the main paper for the sake of brevity as is the one that in average performs better, as observed by [32,44].
> We report comparison with other metalearning baseline in Appendix D.8. The 10% improvement on MiniImagenet is with respect to the method of [47], we will clarify this misunderstanding.
> We respectfully remark that the extensive experiments and evaluation were pointed out by reviewers (BjS9, Aoq2,v gpp) and that we agree that our main claim is to show how sparse sufficiency and minimality are useful properties to achieve disentanglement and OOD robustness.
>
> ### **Minimality and feature sharing**
>
> The minimality principle states that for a task $t$ there does not exist a different assignment of its support (S’) that is able to solve $t$. This means that there are no duplicated features. This is needed when the dimensionality of the representations is higher than the number of factors of variation (which is typically our setting).
> In this case sufficiency is not enough to attain disentanglement as features are still allowed to be duplicated or splitted across the representation entries (see Figure 3 and 6).
>
> **Clustering**: The  regularization minimizes the entropy of the task-averaged distribution $\tilde{\phi}_m$ of feature importances (i.e. absolute value of the coefficients of the linear predictor).  By doing this we enforce  $\tilde{\phi}_m$ to be more  peaked. This means that for a batch of tasks more weight will be given to the more shared features.
> We demonstrate this phenomenon in the paper with the experiment in Figure 6 in Appendix D.1, it can be observed how imposing growing values of regularization strength $\beta$ promotes clustering in the task/feature space, i.e. many tasks will share the same features. When enforced too much this clustering effect collapses everything into one or few features shared among tasks and becomes harmful for performance (see last row of Figure 6).
>
> We believe that this comment should also address the counter example (for which M=1 is an odd assumption as no disentangled representation is possible with continuous factors as per our theory). We’re happy to further discuss these points if needed. Concerning the the scaling factor $\frac{1}{TC}$ under Equation 3, it should appear both at the numerator and denominator of  $\tilde{\phi}_m$. We will fix this, thanks.
>
>
> ### **Waterbirds dataset task generation**
>
> Please see the response to this point in the General Response section.
>
> ### **Regularization at test time**
>
> For a fair comparison we adopt a linear head with the sparsity regularization for both our method and the ERM. We’ll clarify this in Appendix C.4
> The feature sharing is turned off at test time. The only exception are the interesting ablation experiments in D.9. There, we show that using feature sharing at test time we can trade off better performance OOD with slightly lower performances closer to the training distribution (Tab 16, last row). As confirmed by these preliminary experiments, tuning feature sharing at test time on unsupervised test data is an interesting direction for future work.
>
> ### **Finetuning on domain bed**
> For all the experiments on domain shifts, we started with a pretrained network on Imagenet and we finetune it on the target dataset using our method. The networks in the synthetic experiments are instead trained from scratch. In this regard, it would be interesting as future work to train a network from scratch on a large scale dataset like Imagenet with our method, and then testing the transferability of the resulting model compared with ERM pretrained models.
>
> ### **On the necessity of metalearning**
> Please see the response to this point in the General Response section.

---

> > ### Comment · Reviewer_S5sy · 2023-08-15
> >
> > I thank the authors for their response and for clarifying some of the questions. I believe including these clarifications and more baselines presented only in the appendix will improve the work. I am currently reconsidering my grade, and i have the following questions.
> >
> > ### Task generation
> > Unlike mentioned in the general response, the task definition in L107 does not incorporate $U$ and $S$, but only the set of classes $C$. This needs to be defined more concretely. Also, the implication of this definition should be clarified as it differs from the standard definition from few-shot/meta-learning literature, where a task is defined solely by $C$ and different $U$s are just different instantiations of the same task.
> >
> > In addition, the definition in L107 suggests that all $U$s sampled for the same $t$ should be regarded as the same task when calculating the regularizer. However, the authors' response suggests that this is not the case. How does it affect the regularizer?
> >
> > ### Minimality and feature sharing
> > I think my comment "with logits for simplification, i.e., M=1" was confusing and did not align with the paper's definitions. What i meant is that instead of having two logits, one for each class, followed by softmax, in this example, i use a single logit and a sigmoid function for simplicity (which has a one-to-one correspondence to the "full" softmax case). So, as the review says in "For example, consider 3 FoVs $f_i$ [...]", there are 3 features (FoVs), so $M=3$ in this case. I hope this clarifies my example.
> >
> > I do not see how the authors' response addresses this example, where a solution satisfying the feature sharing principle (here, each feature is shared among 2 out of 3 tasks) has the highest entropy. It would be helpful if the authors could clarify this more explicitly.

---

> > > ### Author Response · Authors · 2023-08-16
> > > **Response to Reviewer's comment**
> > >
> > > We thank the reviewer for engaging in the discussion and for their questions. We will address them below:
> > >
> > > ### Task generation
> > >
> > > - We thank the reviewer for the feedback. We will make the definition of task in L107 more precise: we will incorporate the dependance on $U$ and $S$ more explicitly, as well as the suggested considerations and pointers to Appendices C.3 and C.4 where the task generation process is described more in depth.
> > >
> > > - Concerning the question: the reviewer is correct in saying that a task is entirely defined by the samples $U$, $Q$, and the classes $C$ .
> > > This indicates that despite depending on the same classes, two tasks can differ due to variations in their sets of $U$ (and $Q$).
> > > As emphasized in the [General Response](https://openreview.net/forum?id=IHR83ufYPy&noteId=fdPZrYeKDC), by keeping the cardinalities of  $U$ and $Q$ small (i.e. solving few-shot tasks), tasks may exhibit spurious correlation (for instance, in Waterbirds, background and foreground (label) information),  but they may also be free of them, given the small sample size (e.g.for Waterbirds,  the tasks is composed mostly on samples on land background or vice versa).
> > > The regularizers come into play to identify solutions that neglect these spurious correlations. To illustrate this, consider a scenario with a batch containing only two tasks, both predictingreliant on the same classes. If one task retains a spurious correlation, while the other doesn't, **the feature sharing regularizer (combined with the cross entropy loss) prioritizes the solution without the spurious correlation**.
> > > This stems from the fact that enforcing non-spurious features to be shared doesn't harm performance, whereas sharing spurious features negatively impacts the task lacking such correlations.
> > >
> > >
> > > ### Minimality and feature sharing
> > > We thank the Reviewer for clarifying the setting of the suggested example. According to our understanding, the setting is:
> > >
> > > |   	| _f1_ | _f2_ | _f3_ |
> > > |-------|------|------|------|
> > > | _t1_  | x	| x	|  	|
> > > | _t2_  | x	|  	| x	|
> > > | _t3_  |  	| x	| x	|
> > >
> > > where the "x" highlights the dependency between FoVs and tasks and each feature is shared among 2 out of 3 tasks, with $M=3$.
> > >
> > > In this specific case, **given the additional necessary assumptions**, sparsity can be indeed enough to achieve identifiability, as proved in [47] and in our case in Appendix A.
> > >
> > > **Minimality is however satisfied**, as the principle does not states that minimal entropy solutions are the right ones, but that no duplicate features should be used  (i.e. for the characterization in terms of entropy, see Appendix A in the paper).
> > >
> > > The feature sharing regularizer is **one way** to enforce minimality in our setting. This is to address more realistic cases, when sparsity is not enough. One evident example is when the dimensionality of the representation $M$ is higher than the number of factors of variations ( $M>=3$, in the previous example). In such settings duplicated features are much more likely to occur and sparsity alone cannot achieve disentanglement (as further confirmed by the experimental evidence in Figure 5).
> > > **Feature sharing is particularly effective when the number of FoVs is unknown**, which is a more realistic assumption and is more tailored to scale to real data, where labels on the FoVs are unlikely to be available.
> > > As highlighted in the limitation section of the paper, enforcing features to be shared too much,  leads to solutions that don’t satisfy minimality anymore and can lead to degraded performance (see Figure 5, when $\beta >0.25$). The same is valid for the provided example, when $M=3$.  In this regard, good strategies to tune the feature sharing regularization strength are promising future work directions, as we observed in the paper.
> > >
> > >
> > >
> > > We hope to have fully clarified the Reviewer concerns and remain available to clarify any further questions.

---

> > > > ### Comment · Reviewer_S5sy · 2023-08-17
> > > >
> > > > I thank the authors for answering my questions and providing additional clarifications. I am increasing my grade assuming the authors address the clarity and provide mentioned discussions in the camera ready.

---

> > > > > ### Author Response · Authors · 2023-08-18
> > > > > **Response to Reviewer's comment**
> > > > >
> > > > > We thank again the Reviewer for all their suggestions to improve the paper and for raising their score. We commit to address the clarity concerns and incorporate the discussions in the camera ready.

---

### Official Review · Reviewer_vgpp · 2023-06-30

**Soundness:** 3 good
**Presentation:** 3 good
**Contribution:** 3 good
**Rating:** 7
**Confidence:** 4

**Summary:**

The paper studies the problem of learning disentangled representations in a setting with access to multiple supervised tasks, each of which is assumed to depend only on a subset of the underlying factors of variation. While the factors of variation are never directly observed, two principles are incorporated to allow their identifiability; i) sparse sufficiency (“sparsity”) ii) minimality - encouraging features to be shared / reused across tasks. The proposed approach is demonstrated to achieve a higher degree of disentanglement on synthetic data compared to previous methods, as well as better out-of-distribution generalization in real image domains.

**Strengths:**

1. The idea of studying disentanglement in the context of generalization to downstream tasks is well motivated, especially in real image domains.

2. While the “sparsity” regularization already introduced for disentanglement in previous works, including the concurrent work of [47], the second “minimality” term seems novel. Including this term is well justified for real world problems when the number of FoV is unknown, and the results in the paper highlight its effectiveness.

3. The experimental section contains both quantitative disentanglement evaluation on synthetic datasets, as well as appropriate assessment for domain generalization in real image data, where the ground truth for the factors of variation is unknown.


**Weaknesses:**

1. The idea of multitask meta-learning still looks a bit “over” complicated to me. I think the paper would benefit from including the following simple baseline for comparison: combining all the supervision from all tasks into a single multilabel classification task. The feature layer before the classification head should still include the sparsity term. Can the authors elaborate on the ERM baseline? Is this one the “simple” baseline I just mentioned?

2. The dataset considered in [47] is 3Dshapes. However, I do not see a comparison on this benchmark against [47]. Could the authors report the DCI evaluation of the proposed method and [47] on 3Dshapes?


**Questions:**

1. There are several generative approaches to learning disentangled representations which are worth mentioning e.g. [1, 2]. However, these methods rely on weak supervision e.g. observing some of the factors of variation for some of the images. Given the binary labels constructed in this paper for some of the dataset, I wonder if there is a way to “quantify” the amount of supervision provided to the proposed model. This is not trivial as no direct labels are used but as long as many of “binary” labels are constructed the effective level of supervision is increased. Could the authors elaborate on this perspective?

[1] Locatello et al. “Disentangling factors of variations using few labels”. In ICLR, 2020.

[2] Gabbay et al. “An Image is Worth More Than a Thousand Words: Towards Disentanglement in the Wild”. In NeurIPS 2021.

2. DCI metric is a combination of three different metrics: Disentanglement, Completeness and Informativeness. However, the results in this paper are presented as a single number. Is this the average of the three? or just the Disentanglement? I request the authors to present the metrics separately in the quantitative evaluation as they may shed more light on the performance of each of the considered methods.

3. Minor typos:
* Line 45 - w.r.t to
* Line 57 - “cohexist” → “coexist”


**Limitations:**

The authors have adequately addressed the limitations and potential negative societal impact of their work.

---

> ### Author Rebuttal · Authors · 2023-08-10
>
> We thank the reviewer for all their suggestions, comments and questions. We will address them in the following. We remain available during the discussion phase to further clarify any question or concern.
>
> ### **Metalearning and baselines**
>
> For comparison with the ERM baseline we detach the original head (last layer) from the architecture and attach the same sparse linear head that we use in our method to be trained on the support tasks at test time.
> This corresponds to comparing the feature space obtained from our method with the feature space obtained by concatenating all tasks/domains together and training with ERM. For further comments on the choice of employing meta learning in our approach, please see the related point in the General Response section.
>
> ### **Comparison with [47]**
>
> The dataset employed in [47] is indeed 3D shapes but a direct comparison with [47] is not feasible as we don’t have access to the task distribution with which the experiment was performed.
> Nevertheless [47] is akin to our method without the feature sharing regularizer and we show direct comparison with it in Figure 5 (purple curve vs yellow curve). Here we show that when the number of factors of variations is unknown, sparsity alone (yellow dashed line) cannot achieve full disentanglement, in confirmation of our theoretical result in Proposition 2.1.
> **In Figure A of the attached PDF to the General Response, we report the same experiment performed on the 3DShapes dataset, which demonstrates that sparsity alone [akin to 47] is not sufficient to achieve maximal disentanglement**, in this setting.
>
> ### **Supervision on the FOVs**
>
> We consider our method in a similar setting as the weakly supervised method like [58,37,25,a,b]. The supervision in our case comes indirectly from the task label and is necessary for obtaining identifiability results (without any supervision this was proven not possible, e.g. in  [56]).
> Quantifying the amount of supervision given by the task labels, is indeed an interesting direction. Overall we agree with the reviewer that this is not easy to estimate quantitatively this amount due to large variation of unobserved quantities.
> A possible direction would be to inspect the task distribution (assuming full knowledge of it) and estimate the probabilities needed for identifiability in Proposition 2.1, i.e. the assumptions on the task distribution $p(S \cap S' = \{i\})> 0$ or  $p( \{i\} \in (S \cup S') -(S'\cap S))>0$. Namely the probability of two tasks to share a single factor and the probability that two arbitrary tasks share most factors but differ in one.
> A related source of complexity comes from the number of factors of variations (FoVs) in the dataset and the number of possible values (assuming discrete variables) that they can assume.  As this grows, one can expect that more observations from p(x,y) are needed in order to learn disentangled representations, but also that tit becomes harder to estimate the probabilities stated above. Assuming a good estimate of the these probabilities, the number of FoVs, the number of possible values and the probability distribution used to sample them (in  most cases this is uniform), one could draw a loose estimate of the number of observation from p(x,y) needed in practice to learn a representations disentangled w.r.t. to the factors of variation.
>
>
>
> _[a] Locatello et al. “Disentangling factors of variations using few labels”. In ICLR, 2020._
>
> _[b] Gabbay et al. “An Image is Worth More Than a Thousand Words: Towards Disentanglement in the Wild”. In NeurIPS 2021._
>
>
> ### **DCI Metric**
>
> Please see the response to this point in the General Response section.

---

> > ### Comment · Reviewer_vgpp · 2023-08-13
> > **Reviewer's Response to Rebuttal**
> >
> > Thank you for the clarifications.
> >
> > Regarding the comparison with [47] on 3Dshapes, could you please report all DCI components separately as well?
> > (Minor comment: Note that in the PDF rebuttal page the results in Table B are flipped between w/ and w/o regularization)

---

> > > ### Author Response · Authors · 2023-08-14
> > > **Response to reviewer's comment**
> > >
> > >
> > > - We thank the Reviewer for spotting the flipping typo in Table B, we will report the correct one in the paper.
> > >
> > > - In the Table below, we report Completeness and Informativeness scores for the experimental comparison with [47], relative to Figure A in the rebuttal. We report as well the exact disentanglement scores, to have a complete picture.
> > >
> > >
> > > |                     | _DCI disentanglement_ | _DCI completeness_ | _DCI informativeness_ |
> > > |---------------------|-----------------------|--------------------|-----------------------|
> > > | _No regularization_ | 33.8                  | 32.01              | 92.1                  |
> > > | **$ \beta=0$ (akin to [47])**         | 67.0                  | 59.73              | 93.91                 |
> > > | $ \beta=0.025$      | 74.4                  | 66.08              | 95.49                 |
> > > | $ \beta=0.05$       | 73.78                 | 65.08              | 95.24                 |
> > > | $ \beta=0.1$        | 82.7                  | 70.51              | 94.66                 |
> > > | $ \beta=0.15$       | 81.07                 | 69.41              | 94.21                 |
> > > | $ \beta=0.2$        | 81.45                 | 67.92              | 94.23                 |
> > > | $ \beta=0.25$       | 88.88                 | 76.8               | 95.77                 |
> > > | $ \beta=0.3$        | **95.88**             | **85.0**           | **96.42**             |
> > > | $ \beta=0.35$       | 68.99                 | 63.99              | 94.26                 |
> > > | $ \beta=0.4$        | 55.0                  | 54.18              | 93.26                 |
> > >
> > > Completeness scores follow a similar trend with respect to Disentanglement scores, achieving maximal score at $\beta=0.3$, when both sparse sufficiency and minimality properties are enforced. It goes similarly for Informativeness scores, with values more satured around a high score, probably due to the tasks being relatively simple to solve.
> > >
> > > We remain available for any further questions or clarifications.

---

> > > > ### Comment · Reviewer_vgpp · 2023-08-15
> > > > **Updated score**
> > > >
> > > > Thank you for the detailed response. I have increased my score to 7.

---

> > > > > ### Author Response · Authors · 2023-08-18
> > > > > **Response to Reviewer's comment**
> > > > >
> > > > > We thank the Reviewer for raising their score and for all their suggestions to improve the paper.

---

### Official Review · Reviewer_BjS9 · 2023-07-07

**Soundness:** 3 good
**Presentation:** 3 good
**Contribution:** 3 good
**Rating:** 7
**Confidence:** 3

**Summary:**

The foundation of the proposed work is a form of regularization motivated by the problem of disentanglement.  A supervised task is deconstructed into randomly generated sub-tasks, and the same trained representations are used for all sub-tasks (followed by a linear head specific to each sub-task).  Two regularizers are then introduced on the feature activations: one which encourages sparsity (L1) and one which encourages minimality by penalizing the entropy of the distribution of feature activations.

The disentanglement motivation is such: If we assume each sub-task only requires information from a subset of some ground truth disentangled factors of variation of the data, it is argued that the training procedure and regularization will eventually disentangle the factors of variation completely.  Disentangled factors confer various advantages in representation learning, one of which being increased robustness to shifts in the data distribution because spurious factors are able to be separated from the rest. This advantage forms a large part of the proposed work.  After establishing the intended effect of the regularization on synthetic data where generative factors are known, the focus shifts to real data where domain shift is a challenge.

**Strengths:**

The method is sensible while being nontrivial.  It is presented well, and the experimental results, tackling either idealized disentanglement or practical domain shift, cover a lot of ground.  It is interesting to leverage the ability to decompose label information as a way to condition representations and encourage structural properties of extracted factors of variation.

**Weaknesses:**

There are two perspectives with which to view the introduced regularization and training procedure.  The first is in terms of what it accomplishes with respect to disentanglement in synthetic scenarios, and the second in terms of its utility to address domain shift in real datasets (disregarding disentanglement as unrealistic).  The support for both perspectives could be strengthened.

From the disentanglement perspective, it’s not clear how much everything depends on Y containing full information about the generative factors.  If Y depends on fewer factors than X (as is more often the case), how can Proposition 2.1 hold?  For example, if the original labels (before any task deconstruction) have no “angle” information from DSprites, no sub-task should extract angle information, and even with unlimited samples from p(x,y), the “angle” generative factor will be ignored as spurious.  If Y depends on the factors in an entangled way, e.g. I(xpos;Y)=I(ypos;Y)=0, but I(xpos,ypos;Y)>0, then there is no way to create sub-tasks out of Y that depend on xpos but not ypos.  Even with unlimited samples there should be no disentanglement of the xpos and ypos factors.  Thus it seems that the disentanglement is predicated on having I(Y;Z)=H(Z), ie all factor information in Y; if true, this is quite a severe constraint.  More clarity around this point would be helpful, either through a constructed example on the synthetic dataset or elaboration about what’s expected of Y when discussing Proposition 2.1.

From the real world, pragmatism perspective, the domain shift results are numerous yet somewhat insubstantial.  The comparisons in the main text are predominantly to ERM, which does not seem particularly convincing.  In the research cited to support using ERM as a baseline (Gulrajani and Lopez-Paz 2020), the competitiveness seemed to only come from re-tuning for ERM anew everything pertaining to training and model selection -- but I gather all conditions match for the ERM as the proposed method.  Given two methods, identical except that one has two additional hyperparameters to tune, the one with additional hyperparameters should prevail.  There are many more domain shift results in the Supp, though they are thrown together without much in the way of a unifying perspective.  A significant number of baselines are compared to in App. D4,5, yet the results are presented without exposition.  Curiously, the proposed method is a significant outlier, performing several standard deviations above all fourteen baselines, on a single category of each experiment of D4 (S in CLSV, C in PACS, C in ACPR), yet the performance on other categories is always middling or even the worst.  Can the authors speak to this?

**Questions:**

Can the authors please elaborate on the sub-task generation process for the real datasets?  Take Camelyon17 as an example -- how were sub-tasks generated out of this binary classification problem?

**Limitations:**

Yes

---

> ### Author Rebuttal · Authors · 2023-08-10
>
> We thank the reviewer for all their suggestions, comments and questions. We will address them in the following. We remain available during the discussion phase to further clarify any question or concern.
>
> ### **Disentanglement**
>
> The reviewer is correct in that each task does not depend on all the factors of variation at the same time. Proposition 2.1 has two key assumptions on the task distribution. These are $p(S \cap S' = \{i\})> 0$ or  $p( \{i\} \in (S \cup S') -(S'\cap S))>0$. These states that a factor of variation is identifiable when it is in the intersection of two tasks (more precisely, in the intersection between the two sets indexing the factors of variation relevant for each task) or in the union minus the intersection. Intuitively, it means that there are two tasks that have that factor of variation in common or have several factors in common but only one different. Full identifiability of all factors of variations happen only if the union of all tasks span all the factors. If a factor does not influence any task, it is treated as a nuisance variable that is not learned as in [a]. Future works could explore identifiability of blocks of factors, following the weaker identifiability definition of [b].
>
> _[a] Emergence of Invariance and Disentanglement in Deep Representations, Achille and Soatto, JMLR 2018_
>
> _[b] Self-Supervised Learning with Data Augmentations Provably Isolates Content from Style, Von Kügelgen et al., NeurIPS 2021_
>
>
>
>
> ### **Domain shift**
>
> - **Setup** : We performed the experiments on real datasets putting ourselves in the same exact settings, concerning architecture, augmentations, model selection strategy and evaluation of [32,44]. This was to ensure the fairest possible comparison. We compare with the 13 baselines  in Appendix D4,5 (see point below) other than ERM, which follow the same setting, and their results are reported directly from [32,44]. As highlighted in the main paper, we chose to report mainly ERM  for the sake of conciseness as it is the method that performs best on average, which was confirmed by several results [44,32,c].
> We stress that we don’t claim that our method performs best in all settings (see limitations section 5) , but we observe convincing evidence  that the sparse sufficiency and minimality are useful properties for OOD generalization, in most cases.
>
> _[c] Wiles, O., Gowal, S., Stimberg, F., Alvise-Rebuffi, S., Ktena, I., Dvijotham, K., & Cemgil, T. (2021). A fine-grained analysis on distribution shift_
>
> - **Hyperparameters** : we argue that our method does not correspond to ERM with two additional hyperparameters: In particular we differ in that (i) we learn from a distribution of tasks (ii) we leverage a meta learning approach, and, most importantly, (iii) we incorporate two inductive biases in the learning process, namely sparse sufficiency and minimality. What is shared with the settings of [32,44] is the architectures, augmentations, model selection strategy and evaluation to ensure the fairest comparison.
>
> - **Baselines results in Appendix D4** : In the Tables of the appendix D.4 we accidentally made some mistakes in reporting our results, due to a permutation of the column order with respect to Table 4 in the main paper. In Table D.4.1 we accidentally swapped columns “V” and “S” for our method. In Table D.4.2, we swapped columns “A” and “C”. Similarly, In Table D.4.3 we swapped our results for columns A and C. Please refer to the results in Table 4 of the main paper for the correct order and results for our method.
> We apologize for the confusion, we will correct this in the revision. In light of this, our method performs consistently better or on par with others.In general we hypothesize that we have the best performance when the target domain shares a good amount of information with the intersection of the training domains.This leads to very good gaps in performance when the domain is farther from the training distribution and the transfer is more difficult (e.g. we do 83.5 on “S” in PACS). As remarked in the limitations section, more principled ways to tune sparsity and feature sharing regularization parameters could bring performance even higher. We will incorporate all these considerations into Appendix D4,5.
>
> ### **Task distribution on real datasets**
> Please see the response to this point in the General Response section.

---

> ### Author Response · Authors · 2023-08-18
> **Response to Reviewer BjS9**
>
> We thank once more the Reviewer for their feedback and suggestions.
> We hope to have addressed all the Reviewer questions and concerns in the Rebuttal. Please let us know if the Reviewer is satisfied with the answers provided and/or if any further clarifications are needed. Thank you for your consideration.

---

> > ### Comment · Reviewer_BjS9 · 2023-08-18
> >
> > Thank you for the clarification around the disentanglement and the distinction between ERM.  My few questions and concerns have been satisfactorily addressed, and I will raise my score.

---

### Official Review · Reviewer_Aoq2 · 2023-07-10

**Soundness:** 4 excellent
**Presentation:** 4 excellent
**Contribution:** 4 excellent
**Rating:** 8
**Confidence:** 4

**Summary:**

The paper proposes to learn disentangled representations by leveraging knowledge from a diverse set of supervised task objectives. Specifically, the paper shows that disentangled representations can emerge without ground-truth annotations of the factors of variation under two assumptions: 1) sparse sufficiency, and 2) minimality. Practically, the paper implements a meta-learning objective for representation learning with the sparse sufficiency and minimality constraints. The authors also evaluate the learnt representations for out-of-distribution (OOD) generalization.

**Strengths:**

- The paper is very well written and easy to follow.
- The paper appropriately situates the proposed method among all the related works, including highly related concurrent work.
- The paper has multiple significant contributions that would benefit the research community — it presents the theory as well shows experimentally that appropriate inductive biases can sufficiently recover the factors of variation from the data.
- The disentanglement results are promising. The paper contains ablation studies to show the importance of individual components such as the feature sharing property.
- The proposed method is evaluated for domain generalization and domain shift tasks on six different benchmarks, achieving convincing results.

**Weaknesses:**

In my opinion, the paper does not have any major weaknesses. To me, barring a few grammatical mistakes, it already reads better than a lot of accepted papers. The only thing I could suggest is to make the caption of Figure 1 a bit more descriptive so that it conveys a full picture by itself and doesn’t rely on the main text.

**Questions:**

I think the paper is already in a good and sufficient state to be accepted as it is. I do not have any further questions for the authors.

**Limitations:**

The authors have adequately addressed the limitations of the proposed method.

---

> ### Author Response · Authors · 2023-08-10
> **Response to Aoq2**
>
> We thank the Reviewer for all the positive comments and suggestions. We will include the suggested modifications in the main paper, in particular, we will make sure to polish all captions to make them self-contained.

---

### Author Rebuttal · Authors · 2023-08-10

# General Response

We thank all reviewers for their valuable insights, constructive feedback, and inquiries. In particular, we're glad that all reviewers agree on the value of the contribution, the soundness of the idea, the usefulness of the proposed properties, clarity (reviewers Aoq2,BjS9) and extensive experimental evaluation (reviewers Aoq2,BjS9,vgpp).

In this comprehensive response, we aim to address common concerns and questions raised by multiple reviewers. Subsequently, we will provide direct responses to the specific questions posed by each reviewer. We remain available during the discussion period to clarify any further concern or question





## **DCI metric** ( Reviewers **S5sy** and **vgpp**)

The DCI metric reported refers to the disentanglement component of the DCI metric as standardly adopted in many disentanglement method literature see e.g. [56,58,25]. We measured the completeness and informativeness scores of [22] for the models in Table 1, and we report them in the Table below.  We will incorporate this Table in the Appendix of the manuscript.


|                   		 | _DSprites_ | _3DShapes_ | _SmallNorb_ | _Cars_ |
|----------------------------|------------|------------|-------------|--------|
| **Without regularization**    |   		 |   		 |    		 |   	 |
| _DCI Disentanglement_ 	 | 16.6  	 | 44.4  	 | 16.5   	 | 60.5   |
| _DCI Completeness_		 | 17.5  	 | 39.1  	 | 12.9   	 | 50.8   |
| _DCI Informativeness_ 	 | 88.0  	 | 87.6  	 | 90.5   	 | 95.5   |
|                   		 |   		 |   		 |    		 |   	 |
| **With regularization** |   		 |   		 |    		 |   	 |
| _DCI Disentanglement_ 	 | 69.9  	 | 87.7  	 | 60.5   	 | 92.3   |
| _DCI Completeness_		 | 72.3  	 | 88.4  	 | 63.2   	 | 57.1   |
| _DCI Informativeness_ 	 | 96.0  	 | 95.7  	 | 95.4   	 | 99.7   |


## **Task distribution on real datasets** (Reviewers **BjS9** and **S5sy**)

The procedure is described in Appendix C.4 and we summarize it here:

Each task is a  $C$-class classification problem. To generate it, we sample randomly  $C$ classes from the set of available $K_{train}$ classes in the dataset. For each of the $C$ classes we then sample  $| U |$ support samples to fit the linear head, solving Equation 5, and $| Q |$ query sample to update the backbone weights, optimizing Equation 4. In our experiments, we set $C=5$, when $K_{train}$> $C$, while for the datasets where $K_{train} <5$, we set $C=k_{train}$.
For binary datasets such as Camelyon17 or Waterbirds the possible classes to be predicted are always the same across tasks: what is changing is the composition of $U$ and $S$.
Keeping their cardinality low, we ensure that some tasks will not contain spurious correlation that may be present in the dataset, while other ones will still retain it, and the regularizers will satisfy solutions which discards the spurious information. We can observe evidence of this in the experimental results in Table 4 and in Figure 7 in the Appendix.


## **On the necessity of metalearning**  (Reviewers **S5sy** and **vgpp**)

Meta learning is adopted for two key reasons:
- *Scaling* :  as we’re working with thousands of tasks it would be impractical to keep in memory the classification heads for each task. For example, in the domain shift a lower bound of the estimate of the possible tasks is $\binom{D*K_{train}}{C}$ where $D$ domains, $K_{train}$ classes in each domain, and $C$ classes composing a task.
- *Generalization* :  we employ a meta learning objective to generalize better to unseen tasks. The objective in equation 4 is indeed formulated  w.r.t. to query samples  $Q$, while the linear head is fitted on the support samples $S$ for each task. This is not possible in standard multitask learning formulations.

---

### Decision · Program_Chairs · 2023-09-21

**Decision:**

Accept (spotlight)

**Comment:**

All the reviewers unanimous agree to accept this paper.

 The authors are encouraged to improve this work by addressing the concerns of reviewers in the camera ready, such as

1.	Add experiments on 3Dshapes

2.	Add comparisons with other disentangled representation learning methods

3.	Improve the clarity of the paper